# Microtubules regulate pancreatic β-cell heterogeneity via spatiotemporal control of insulin secretion hot spots

Kathryn P Trogden[1], Justin Lee[1†], Kai M Bracey[1†], Kung-Hsien Ho[1†], Hudson McKinney[1†], Xiaodong Zhu[1,2], Goker Arpag[1], Thomas G Folland[3], Anna B Osipovich[4,5], Mark A Magnuson[1,4,5], Marija Zanic[1,6,7], Guoqiang Gu[1], William R Holmes[8,9,10], Irina Kaverina[1*]

[1]Department of Cell and Developmental Biology and Program in Developmental Biology, Vanderbilt University, Nashville, United States; [2]Department of Medicine, Vanderbilt University, Nashville, United States; [3]Department of Mechanical Engineering, Vanderbilt University, Nashville, United States; [4]Department of Molecular Physiology and Biophysics, Vanderbilt University, Nashville, United States; [5]Center for Stem Cell Biology, Vanderbilt University, Nashville, United States; [6]Department of Chemical and Biomolecular Engineering, Vanderbilt University, Nashville, United States; [7]Department of Biochemistry, Vanderbilt University, Nashville, United States; [8]Department of Physics and Astronomy, Vanderbilt University, Nashville, United States; [9]Department of Mathematics, Vanderbilt University, Nashville, United States; [10]Quantitative Systems Biology Center, Vanderbilt University, Nashville, United States

**\*For correspondence:**
irina.kaverina@vanderbilt.edu

[†]These authors contributed equally to this work

**Competing interest:** The authors declare that no competing interests exist.

**Abstract** Heterogeneity of glucose-stimulated insulin secretion (GSIS) in pancreatic islets is physiologically important but poorly understood. Here, we utilize mouse islets to determine how microtubules (MTs) affect secretion toward the vascular extracellular matrix at single cell and subcellular levels. Our data indicate that MT stability in the β-cell population is heterogenous, and that GSIS is suppressed in cells with highly stable MTs. Consistently, MT hyper-stabilization prevents, and MT depolymerization promotes the capacity of single β-cell for GSIS. Analysis of spatiotemporal patterns of secretion events shows that MT depolymerization activates otherwise dormant β-cells via initiation of secretion clusters (hot spots). MT depolymerization also enhances secretion from individual cells, introducing both additional clusters and scattered events. Interestingly, without MTs, the timing of clustered secretion is dysregulated, extending the first phase of GSIS and causing oversecretion. In contrast, glucose-induced $Ca^{2+}$ influx was not affected by MT depolymerization yet required for secretion under these conditions, indicating that MT-dependent regulation of secretion hot spots acts in parallel with $Ca^{2+}$ signaling. Our findings uncover a novel MT function in tuning insulin secretion hot spots, which leads to accurately measured and timed response to glucose stimuli and promotes functional β-cell heterogeneity.

## Editor's evaluation

The study addresses the functional heterogeneity of glucose-stimulated insulin secretion by single pancreatic β cells. A focused analysis of spatiotemporal patterns of secretion in pancreatic islets led to the identification that cytoskeletal transport networks trigger secretion hot spots at the cell membrane and, in doing so, provide a layer of control over β cell heterogeneity. The study provides a yet uncharacterized dimension to the regulation of insulin secretion from the pancreas.

## Introduction

Insulin secretion in pancreatic β-cells is tightly regulated and highly heterogeneous. The major stimulator for secretion is glucose, with other nutritional and neuronal signals modulating the response. The prevailing view is that glucose influx results in increased glucose metabolism, higher levels of the cytoplasmic ATP/ADP ratio, and increased levels of metabolite intermediates. The increased ATP/ADP ratio results in the closure of $K_{ATP}$ channels, β-cell depolarization, and $Ca^{2+}$ influx that triggers vesicular-cell membrane fusion. Metabolite intermediates promote secretion via known (e.g., microtubule [MT]-dependent vesicular biogenesis, movement, and docking) and unknown mechanisms. Intriguingly, β-cells secrete on the order of tens of granules per cell in response to each round of high-glucose stimulus, which last for hours, despite having thousands of available granules (*Dean, 1973*; *Olofsson et al., 2002*; *Rorsman and Renström, 2003*). Glucose-stimulated insulin secretion (GSIS) has characteristic bi-phasic kinetics with a rapid, high peak within minutes after the stimulation (first phase) followed by a sharp decrease and a slow, lower secreting second phase. The mechanisms underlying this timing are believed to be associated with the availability of secretion-ready insulin vesicles, the pool size of which does not depend on changes in $Ca^{2+}$ levels (*Gaisano, 2017*).

An intriguing property of β-cells is their heterogeneity in morphology, biochemical features, and function (*Miranda et al., 2021* and references therein). Accordingly, β-cells have been found to display different metabolic properties (*Van De Winkel and Pipeleers, 1983*; *Kiekens et al., 1992*; *Van Schravendijk et al., 1992*; *Giordano et al., 1993*), $Ca^{2+}$ influx kinetics (*Zhang et al., 2003*), proliferation rate (*Bader et al., 2016*), stress responses (*Dorrell et al., 2016*; *Johnston et al., 2016*; *Farashi et al., 2018*; *Lei et al., 2018*), expression of some key function- (*Jetton and Magnuson, 1992*) or marker-genes (*Dorrell et al., 2016*), and insulin secretion (*Giordano et al., 1991*; *Van Schravendijk et al., 1992*; *Wojtusciszyn et al., 2008*). Such heterogeneous properties are considered important, because different β-cell subsets could contribute to different physiological needs (*Hoang Do and Thorn, 2015*).

Several mechanisms have been proposed to explain β-cell heterogeneity. For example, it has been proposed that β-cell heterogeneity can result from differences in age, disease state, and location within the islet (*Dean and Matthews, 1970*; *Stefan et al., 1987*; *Ballian and Brunicardi, 2007*; *Aguayo-Mazzucato et al., 2017*; *Gutierrez et al., 2017*; *Pipeleers et al., 2017*). Specifically, β-cell age/disease states can affect gene expression, vesicle maturity, and glucose metabolic flux (*Blum et al., 2012*; *Aguayo-Mazzucato et al., 2017*; *Zeng et al., 2017*; *Huang et al., 2018*). Specific locations in islets can result in differential cross talk with other islet cell subtypes (*Efendić and Luft, 1975*; *Pipeleers et al., 1982*; *Wojtusciszyn et al., 2008*; *van der Meulen et al., 2015*) and with vasculature (*Ballian and Brunicardi, 2007*; *Low et al., 2014*). However, heterogeneous GSIS is maintained in dissociated single cells (*Wojtusciszyn et al., 2008*) and in adults when most β-cells are mature (*Li et al., 2011*; *Hoang Do and Thorn, 2015*; *Dwulet et al., 2021*). In this regard, approximately 25% of adult β-cells remain unresponsive regardless of the glucose concentration (*Hoang Do and Thorn, 2015*). More importantly, islet β-cells show the near-uniform glucose- or depolarization-induced $Ca^{2+}$ influx, despite of the lack of uniform insulin secretion (*Li et al., 2011*; *Dwulet et al., 2021*). These findings indicate that the currently proposed mechanisms cannot fully explain β-cell heterogeneity and are consistent with a possibility that different β-cells may have different numbers of $Ca^{2+}$-responsive insulin granules, contributing to their unique GSIS responses.

In addition to secretory heterogeneity in the β-cell population, uneven distribution of secretion has been also observed at subcellular levels. Designation of insulin secretion to specialized loci at the cell membrane has been shown to be essential for the pathophysiology of type 2 diabetes (*Fu et al., 2019*). The sites of preferential insulin secretion depend on the cellular location of L-type voltage-dependent $Ca^{2+}$ channels (VDCCs) in combination with molecular tethers and other exocytotic proteins (*Bokvist et al., 1995*; *Ohara-Imaizumi et al., 2019a*). This molecular machinery resembles the composition of 'active zones' or areas of high exocytosis in neurons (*Garner et al., 2000*) and is thought to underlie the hot spots of secretion at the plasma membrane (*Landis et al., 1988*). Interestingly, some major components of the hot spot machinery (e.g., ELKS) are assembled at the membrane in response to integrin activation by vascular extracellular matrix (ECM) proteins such as laminin (*Hotta et al., 2010*; *Nishimune, 2012*), and in islets they are preferentially found at sites of β-cell contact with the vasculature (*Ohara-Imaizumi et al., 2005*; *Low et al., 2014*). However, as most β-cells in an in-situ islet have discrete points of contact with the capillaries (*Low et al., 2014*), it is clear that positioning and

ECM-dependent activation of secretion hot spots is insufficient for the differences in secretion activity of individual cells. Thus, despite a significant progress toward the understanding of β-cell heterogeneity, some key cellular mechanisms underling their functional differences in islets are still unknown.

Previously, we have shown that the secretion capacity of β-cells is regulated by the MT cytoskeleton, which is uniquely structured to help tune β-cell function (*Zhu et al., 2015*). Unlike in many other cell types, MTs in β-cells do not radiate from a central point in the cell, and instead form a dense mesh-like network (*Zhu et al., 2015*; *Trogden et al., 2019*; *Bracey et al., 2020*). This network configuration is, to a large extent, due to the fact that most MTs in β-cells originate at the Golgi apparatus (*Zhu et al., 2015*; *Trogden et al., 2019*). The large surface area of the Golgi acts as an MT organizing center (MTOC), leading to an unusual non-radial MT network appearance. Intriguingly, optimal β-cell function depends on a finely tuned balance of MT assembly and disassembly, which can both promote and restrict secretion capacity of a cell. On the one hand, Golgi-derived MTs function aids the budding of new insulin granules from the Golgi (*Trogden et al., 2019*). High glucose stimuli lead to an increase in Golgi-derived MT nucleation, which is necessary to replenish insulin granule content after a secretion pulse (*Zhu et al., 2015*; *Trogden et al., 2019*). Long-term loss of this MT subpopulation causes degranulation of the β-cell as vesicle budding is less efficient (*Trogden et al., 2019*). Thus, glucose-dependent MT nucleation is a critical factor supporting the capacity of β-cells to secrete. On the other hand, MTs act to prevent over-secretion in functional β-cells, which contain excessive amounts of insulin granules (*Zhu et al., 2015*). This function relies on the configuration of MT networks in β-cells, where the non-radial mesh in the cell interior prevents directional granule movement, and the extremely stable MT bundles extending along the plasma membrane serve as tracks for granule withdrawal from docking sites (*Bracey et al., 2020*). Upon glucose stimulation, these pre-existing MTs are destabilized and partially depolymerized (*Ho et al., 2020*), allowing for release of a subset of granules. This regulated dynamicity of the MT network is vital for the dosage of GSIS at each stimulus: loss of all MTs acutely leads to over-secretion, while hyper-stabilization of MTs greatly suppresses it (*Lacy et al., 1972*; *Howell et al., 1982*; *Hill and Rhoten, 1983*; *Zhu et al., 2015*). These effects are only seen after GSIS stimulation but not at basal glucose conditions, indicating the leading role for other mechanisms in secretion triggering (*Zhu et al., 2015*; *Trogden et al., 2019*). In functional β-cells, which are able to secrete in response to a glucose stimulus, the effects of MT destabilization are finely tuned because simultaneous increase in MT nucleation promptly replaces depolymerizing MTs (*Zhu et al., 2015*). The intriguing combination of positive (long-term) and negative (short-term) MT regulation of secretion may be responsible for the early controversy on the role of MTs in insulin secretion (*Lacy et al., 1968*; *Lacy et al., 1972*; *Malaisse et al., 1974*; *Howell et al., 1982*; *Hill and Rhoten, 1983*; *Mourad et al., 2011*). It also makes the MT cytoskeleton a candidate for differential control of secretory activity, which acts in concert with other regulatory mechanisms to underlie functional β-cell heterogeneity.

In this study, we investigate the role of the MT cytoskeleton in β-cells as a factor in the differential secretory response of β-cells to glucose stimulation. Our computational analysis of insulin secretion toward the vascular ECM at the single-cell level in whole islets shows that this regulation occurs via secretion hot spot activity. Our findings indicate that the presence of stable MTs attenuates initiation of secretion hot spots in both otherwise dormant and already active β-cells, thus contributing to functional heterogeneity of insulin secretion. We also show that MTs regulate the timing of insulin secretion by restricting hot spot activity to the first phase of GSIS.

## Results

### Microtubule stability in pancreatic islet β-cells is heterogenous

We have previously found that MTs serve as a critical regulator of both phases of β-cell glucose response and insulin secretion (*Zhu et al., 2015*; *Trogden et al., 2019*). To test whether MT regulation plays a role in the functional heterogeneity of β-cell population, we analyzed MT stability in mildly disseminated mouse islet β-cells plated on vascular ECM.

First, MT stability was evaluated utilizing immunostaining for Glu-tubulin (also known as detyrosinated tubulin), a marker of long-lived or stable MTs (*Gundersen et al., 1987*; *Wehland and Weber, 1987*; *Figure 1A–F*, β-cells outlined in yellow). β-cells were identified by expression of nuclear-localized mApple marker (see Materials and methods). Consistent with our previous findings, Glu-tubulin

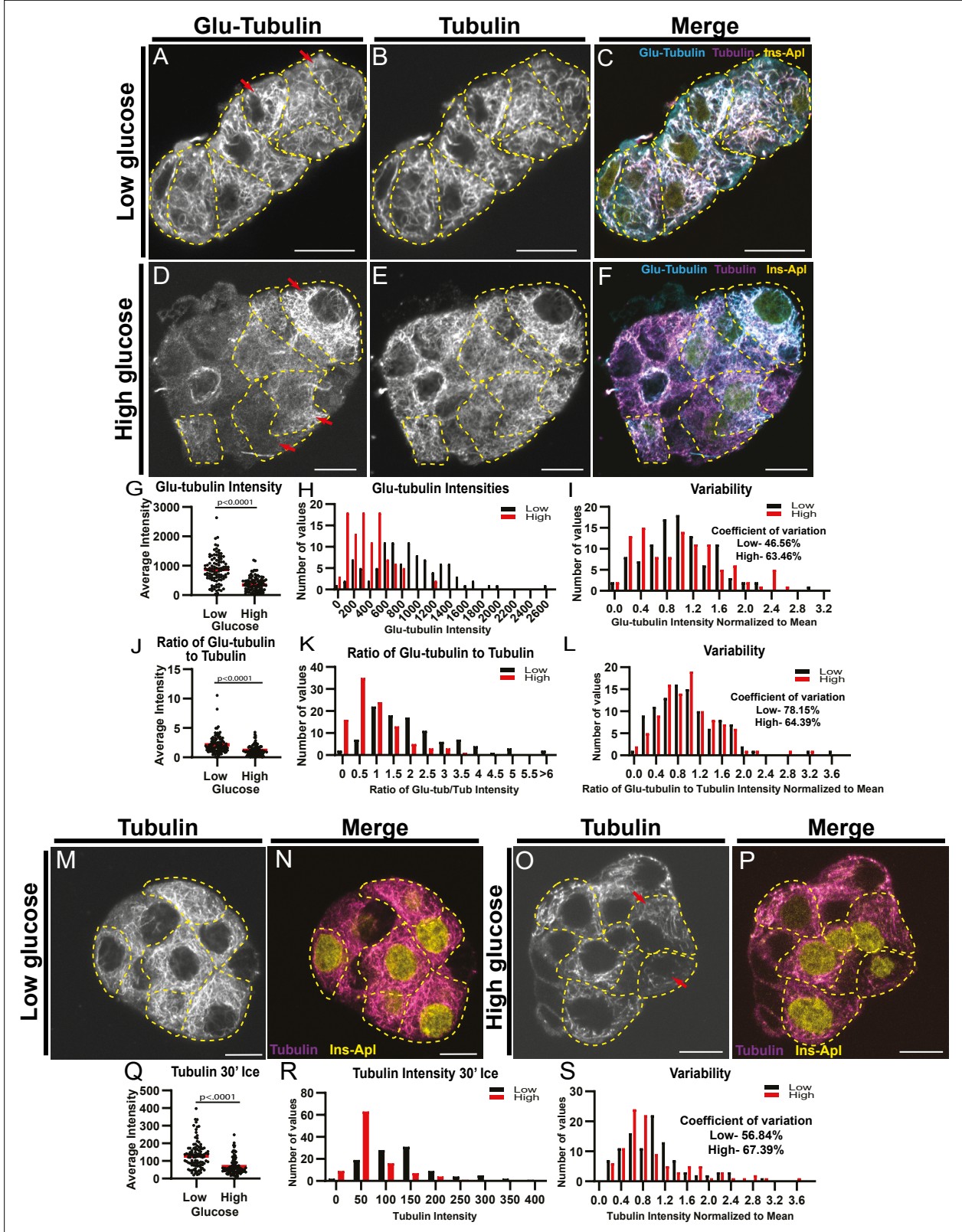

**Figure 1.** MT stability decreases in high glucose but remains heterogenous. (**A–F**) Disseminated islets treated with low (**A–C**) and high (**D–F**) glucose stained for Glu-tubulin (**A, D**) and tubulin (**B, E**). β-cells (dashed yellow line) were identified using Ins-Apl red nuclei (yellow, **C, F**). Red arrows point to differences between cells. Merge (**C, F**) shows Glu-tubulin (cyan), tubulin (magenta), and red nuclear expression of Ins-Apl (yellow). Single slice from the bottom of the cells. Scale bars: 10 μm. (**G**) Scatterplot of Glu-tubulin average intensity for each cell. Mean, red bar. Student's t-test, p<0.0001.

*Figure 1 continued on next page*

*Figure 1 continued*

n=101 cells per condition. (**H**) Histogram of Glu-tubulin average intensity in low (black) and high (red) glucose. Bin=100. n=101 cells per condition. (**I**) Histogram of Glu-tubulin average intensity normalized to the mean of each low (black) and high (glucose). Bin=0.2. Coefficient of variation=standard deviation/mean. n=101 cells per condition. (**J**) Scatterplot of Glu-tubulin to tubulin ratio of average intensity for each cell. Mean, red bar. Student's t-test, p<0.0001. n=101 cells per condition. (**K**) Histogram of Glu-tubulin to tubulin ratio of average intensity in low (black) and high (red) glucose. Bin=0.5, overflow bin of >6. n=101 cells per condition. (**L**) Histogram of Glu-tubulin to tubulin ratio of average intensity normalized to the mean of each low (black) and high (glucose). Bin=0.2. Coefficient of variation=standard deviation/mean. n=101 cells per condition. (**M–O**) Disseminated islets placed on ice for 30 min in low (**M, N**) and high glucose (**O, P**) and stained for tubulin (**M, O**). β-cells (dashed yellow line) were identified using red nuclear expression of Ins-Apl (yellow, **N, P**), merged with tubulin (magenta **N, P**). Red arrows point to differences between cells. Single slice from the bottom of the cells. Scale bars: 10 µm. (**Q**) Scatterplot of tubulin average intensity for each cell after 30 min in high glucose. Mean, red bar. Student's t-test, p<0.0001. n=100–101 cells per condition. (**R**) Histogram of tubulin average intensity in low (black) and high (red) glucose after 30 min on ice. Bin=50. n=100–101 cells per condition. (**S**) Histogram of tubulin average intensity normalized to the mean of each low (black) and high (glucose) after 30 min on ice. Bin=0.2. Coefficient of variation=standard deviation/mean. n=100–101 cells per condition. MT, microtubule.

The online version of this article includes the following figure supplement(s) for figure 1:

**Source data 1.** Data for graphs depicted in *Figure 1G,H,J,K,L,Q,R,S* and *Figure 1A,B,C,F,G,H,I,J,K*.

**Figure supplement 1.** MT stability is regulated by glucose stimulation.

---

content was high in β-cells at basal glucose conditions, but decreased following a high glucose stimulus (*Zhu et al., 2015*; *Ho et al., 2020*; *Figure 1G and H*). There was no detectable difference in overall polymerized tubulin content upon glucose stimulus (*Figure 1—figure supplement 1A*), likely due to glucose-stimulated nucleation of new MTs (*Trogden et al., 2019*), which replace destabilized MTs (*Zhu et al., 2015*; *Trogden et al., 2019*). Accordingly, the ratio of Glu-tubulin to tubulin per cell, which reflects the proportion of stable MTs within the MT network, decreased after high glucose stimulation (*Figure 1J and K*).

Interestingly, the density of the MT cytoskeleton was heterogeneous within the β-cell population. While not changed by glucose stimulation, the overall tubulin polymer content was variable in the β-cell population as indicated by the intensity distributions (*Figure 1—figure supplement 1B and C*) and coefficients of variation (58.72% in low glucose and 60.42% in high glucose). MT stability was also highly variable, as evident from the Glu-tubulin staining, which ranged from barely detectable levels to highlighting essentially the whole MT cytoskeleton (*Figure 1A and D*, compare cells with red arrows). While the amount of Glu-tubulin decreased after glucose stimulation, shifting the histogram of Glu-tubulin intensities per cell to the left (*Figure 1H*), the degree of variation amongst the β-cells was retained as indicated by the distributions of Glu-tubulin intensity normalized to the mean for each condition (*Figure 1I*), and by high coefficients of variation (48.56% in low glucose and 63.46% in high glucose). The proportion of stable MTs within the MT network (Glu-tubulin to tubulin) also remained variable in high glucose (*Figure 1K and L*; coefficient of variation: 78.15% in low glucose and 64.39% in high glucose).

As a second measure of MT stability in β-cells, we subjected the cells to ice treatment for 30 min. Since MTs are temperature-sensitive and only stable MTs will remain after ice treatment, this well-established assay is used to directly assess MT stability. After extraction of free tubulin, fixation, and immunostaining, the tubulin content per cell was measured to evaluate the amount of cold-resistant MTs remaining (*Figure 1M–P*). In basal glucose conditions, many cells contained MTs stable enough to be retained after ice treatment (*Figure 1M and N*). Following high glucose stimulation, the amount of cold-stable MTs significantly decreased (*Figure 1O,P,Q,R*), consistent with our previous findings and data described above (*Zhu et al., 2015*). Most of the remaining MTs were positive for Glu-tubulin, as expected for long-lived stable MTs (*Figure 1—figure supplement 1D-K*). Importantly, the variation in cold-resistant MTs within the β-cell population was high both in low and high glucose (*Figure 1S*), confirming that MT stability was highly variable in the β-cell population.

As a third test of MT stability, we utilized a method where live cells were permeabilized prior to fixation to release all free tubulin, which acutely decreases tubulin concentration in the cell (*Khawaja et al., 1988*). This prevents tubulin polymerization so that dynamic MTs are lost through depolymerization without being replaced, leaving only stable MTs in a cell (*Khawaja et al., 1988*). Immunofluorescent detection of MTs after this treatment resulted in a similar variation in MTs patterns retained in cells and Glu-tubulin staining as the cold treatment (*Figure 1—figure supplement 1L-N*).

These three assays indicated that, similar to insulin secretion, MT stability is heterogeneous between different β-cells. While some cells have highly stable MT networks, neighboring cells can have less stable networks. This phenomenon is even more obvious after high glucose stimulation when MT stability is decreased in the entire population, but remains highly variable.

## MT stability affects insulin secretion heterogeneity

To assess whether MT regulation affects the functional heterogeneity of β-cells, we next analyzed insulin secretion from distinct cells within the β-cell population. To detect single-cell insulin secretion in real time, we used a cell impermeant zinc dye FluoZin-3 that becomes fluorescent upon zinc binding, and at each exocytic event, highlights the release of zinc co-packaged with insulin in granules (*Gee et al., 2002*; *Zhu et al., 2015*). Intact islets were attached to a vascular ECM, a condition that imitates secretion toward vasculature (*Bonner-Weir, 1988*; *Low et al., 2014*; *Gan et al., 2018*) and preserves the ability of islets to efficiently respond to glucose stimulus (*Patterson et al., 2000*; *Zhu et al., 2015*). Total internal reflection fluorescent (TIRF) microscopy was used to analyze secretion from islet β-cells at the attached islet side (See Materials and methods, *Figure 2—figure supplement 1A*). The advantage of this model is that it allows for detecting secretion distribution within two dimensions of plasma membrane contacting vascular ECM (a model for the cell area contacting capillaries). Individual secretion events appear as flashes of bright fluorescence, the intensity of this fluorescence over time resembles a Gaussian curve with only background signal, then an initially tight circle of signal that dissipates after vesicle exocytosis (*Figure 2A and A'*). For quantification, image sequences were processed for better signal/noise ratio, β-cells were identified by the presence of red fluorescence in the nuclei, and cell outlines were detected by bright-field imaging.

To address the heterogeneity of β-cell activity in our experimental setup, we first measured the percentage of β-cells secreting within 10 min of stimulation. In control islets treated with high glucose, the number of active cells increased as expected (*Figure 2B and E*, *Figure 2—figure supplement 1B* and *Figure 2—video 1*; *Low et al., 2013*). However, a subset of β-cells remained inactive even after stimulation (*Figure 2E*, 60% of cells), resembling the fraction of cells with a high content of stable MTs (59% of cells with >300 average intensity of Glu-tubulin, *Figure 1*).

To assess the role of MTs in the distribution of secretion activity in the β-cell population, we preincubated islets in the presence of either the MT depolymerizing drug nocodazole, which completely eliminates cellular MTs (*Figure 2—figure supplement 1F*), or the MT-stabilizing drug taxol (*Figure 2—figure supplement 1G*), which hyper-stabilizes MT networks in cells, dramatically increasing their Glu-tubulin content (*Gundersen et al., 1987*; *Wehland and Weber, 1987*; *Townley et al., 2015*). As we have previously observed (*Zhu et al., 2015*), nocodazole increased insulin secretion specifically in high glucose-stimulated islets (*Figure 2C and E*, *Figure 2—figure supplement 1C* and *Figure 2—video 2*). Strikingly, we also observed that a larger sub-population of β-cells (on average, 66% of cells) was activated, compared to only 42% in control islets (*Figure 2E*, compare *Figure 2B and C*). Addition of taxol also affected insulin secretion as we have previously observed (*Zhu et al., 2015*), blunting it in high glucose (*Figure 2D and E*, *Figure 2—figure supplement 1D*, *Figure 2—video 3*). The percent of secreting cells in taxol was only 27%, similar to the percentage observed in low glucose (*Figure 2D and E*, *Figure 2—figure supplement 1D*), where MTs are intrinsically stable (*Figure 1*).

These results indicate that the modulation of MT stability can change the insulin secretion activity of β-cells within the population. Since in control islets, cells have variable MT stability levels (*Figure 1*), we explored a potential connection between MT stability in individual cells and their ability to secrete. This connection was assessed by a correlative microscopy approach using whole-mount attached islets (*Figure 2F–H*, *Figure 2—figure supplement 2*) as well as slightly disseminated islets (*Figure 2—figure supplement 2B-F*). FluoZin-3-detected secretion pattern was correlated with post-assay Glu-tubulin immunostaining. As expected, secretion levels were extremely low in cells with a high content of Glu-tubulin (Glu-tubulin intensity above islet average), indicating that cells with overly stable MTs are not capable of secretion (*Figure 2H*, *Figure 2—figure supplement 2A-C*). Interestingly, in cells with Glu-tubulin content below the islet average, secretion levels were highly variable, indicating that in those cells other, non-MT-dependent mechanisms significantly modulate GSIS levels.

We next tested whether MT presence introduces differences between actively secreting β-cells by assessing how modulation of MTs influences the number of secretion events per cell, a criterion critical for GSIS efficiency (*Low et al., 2013*). We found that high glucose caused an increase in the

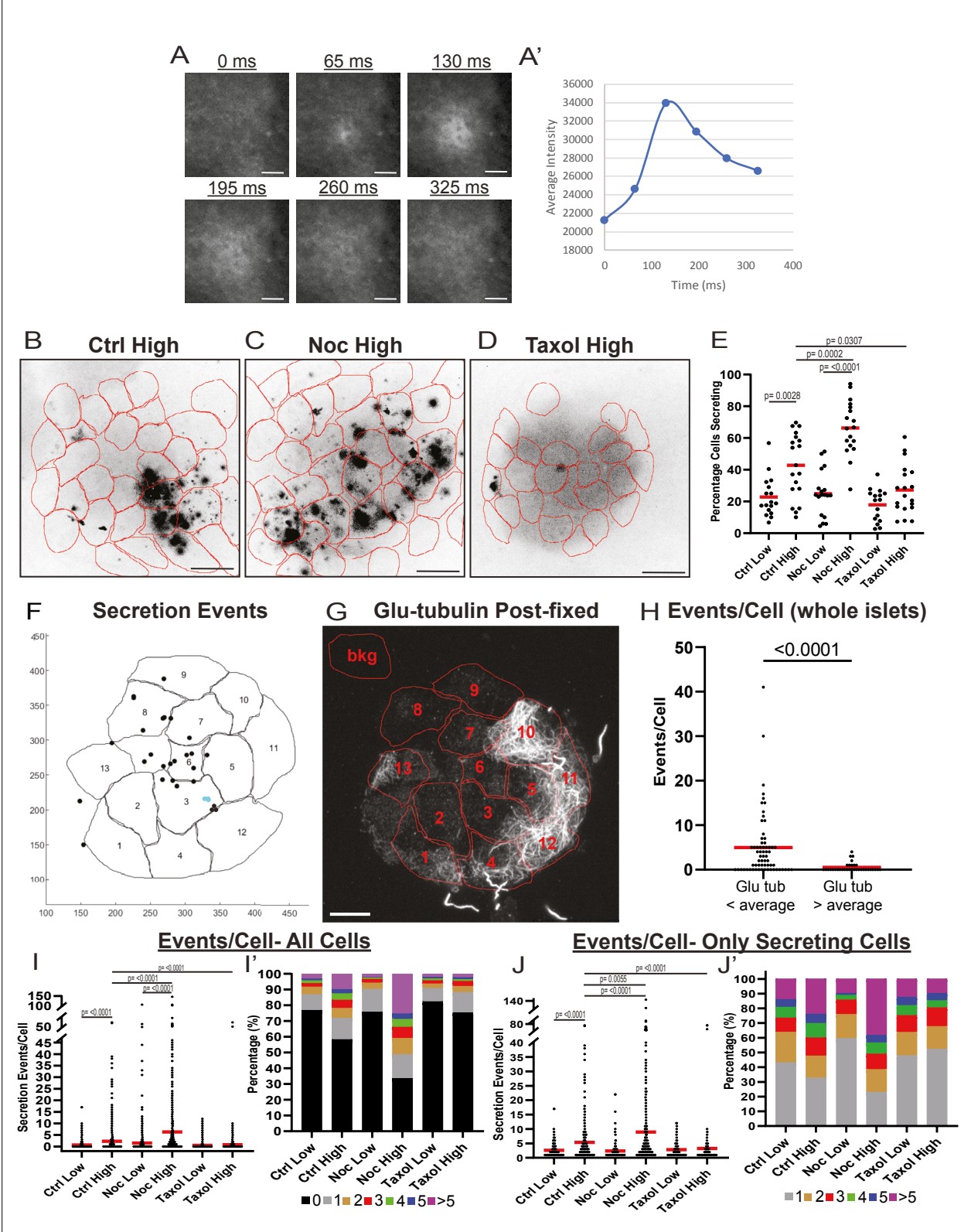

**Figure 2.** MT instability increases β-cell activation and insulin secretion. (**A**) Example images of a single secretion event pre-processing. Secretion signal starts at 65 ms and dissipates out. (**A'**) Graph of the average intensities of a circular ROI from images in panel (A). (**B–D**) Time projections of islets from *Figure 2—videos 1–3* inverted. Fluozin-3 flashes are represented as black areas. Cell borders identified via pre-assay imaging (see Materials and methods) overlaid in red. Islets were preincubated in DMSO (control, B), nocodazole (C), or taxol (D) and stimulated with 20 mM glucose. Scale bars:

*Figure 2 continued on next page*

*Figure 2 continued*

100 µm. (**E**) Graph of the percentage of cells in each field of view with at least one secretion event. Red bars, mean. One-way ANOVA and multiple comparison tests, p-value as indicated. N=16–19 islets. Here and below, islets derived from three or more independent isolations per each condition. (**F**) Glucose-stimulated secretion as detected by FluoZin-3 assay in islets. A representative output image from Matlab script (see Materials and methods) shows cell outlines (black lines) and secretion events (dots). Black dots are non-clustered secretion events, colored dots are clustered secretion events. (**G**) The same islet as in (**F**) fixed after the assay and stained for Glu-tubulin (grayscale). Cell outlines within which Glu-tubulin intensity was measured are shown in red. Numbers correspond to cells in (**F**) with the same number. The outline 'bkg' indicates background measurement area. Maximum intensity projection over 1.2 µm at the bottom of the islet. Bar: 10 µm. (**H**) Correlative analysis between data as in (**F**) and (**G**). The number of secretion events per cell with Glu-tubulin intensity below islet average and those above islet average is compared in the graph. Mann-Whitney nonparametric comparison test p-value is shown. N=98 cells from five islets. The same data set as in *Figure 2—figure supplement 2A*. (**I**) Graph of secretion events per cell detected by FluoZin-3 assay. All cells in a field of view are analyzed, whether activated during the movie or not. Red bar, mean. Kruskal-Wallis test nonparametric and multiple comparison tests, p-value as indicated, N=495–637 cells from 16 to 19 islets. (**I'**) Cells from panel (**H**), graphed as a stacked histogram of the percentage of total cells per condition that had each number of secretion events. (**J**) Graph of secretion events per cell only including cells with at least one event during the duration of the movie. Red bar, mean. Kruskal-Wallis test nonparametric and multiple comparison tests, p-value as indicated, N=88–407 cells from 16 to 19 islets. (**J'**) Cells from panel (**I**), graphed as a stacked histogram of the percentage of cells that had each number of secretion events. MT, microtubule; ROI, region of interest.

The online version of this article includes the following video and figure supplement(s) for figure 2:

**Source data 1.** Data for graphs depicted in *Figure 2E, H,I,I',J,J'* and *Figure 2—figure supplement 2A, B, C*.

**Source data 2.** Matlab script to identify secretion events and clusters.

**Source data 3.** Matlab script to identify secretion events and clusters for correlative microscopy.

**Figure supplement 1.** Assay protocol and basal glucose conditions are not affected by MT stability.

**Figure supplement 2.** Correlation of insulin secretion and MT stability.

**Figure 2—video 1.** Control islet insulin secretion in low and high glucose.

https://elifesciences.org/articles/59912/figures#fig2video1

**Figure 2—video 2.** Nocodazole islet insulin secretion in low and high glucose.

https://elifesciences.org/articles/59912/figures#fig2video2

**Figure 2—video 3.** Taxol islet insulin secretion in low and high glucose.

https://elifesciences.org/articles/59912/figures#fig2video3

number of events per cell in both control and nocodazole-treated islets, but not taxol-treated islets (*Figure 2I-J'*). Importantly, the number of events per cell and the fraction of highly-secreting cells in high glucose were significantly increased by nocodazole as compared to control, suggesting that MT presence attenuates glucose-stimulated secretion in individual cells (*Figure 2I-J'*). Consistent with this result, under all conditions of high MT stability (low glucose in all conditions and high glucose in taxol) secretion activity is distributed similarly throughout the cell population (*Figure 2I' and J'*). Taken together, these data indicate that insulin secretion capacity on both the cellular and population levels is decreased by MT presence, and even more so, by MT stability. Furthermore, in terms of β-cell activation, MTs provide a mechanism that supports β-cell heterogeneity. While normally the glucose-stimulated β-cell population is highly heterogeneous, changing MT stability in either direction leads to a more homogenous population with either most cells secreting (nocodazole, no MTs) or most cells not secreting (taxol, hyper-stabilized MTs). It is also evident that modulation of MT dynamics decreases but does not completely eliminate variability in β-cell activity, which is in agreement with the existence of other, MT-independent, mechanisms of heterogeneity (reviewed in *Gutierrez et al., 2017*; *Pipeleers et al., 2017*).

## MT stability regulates β-cell GSIS by suppressing insulin secretion "hot spots"

To better understand MT-dependent regulation of secretion in individual cells versus a cell population, we analyzed their spatial and temporal secretion patterns. In agreement with the existing evidence that insulin secretion occurs in hot spots, defined by patches of VDCCs and other active zone proteins on the membrane (*Ohara-Imaizumi et al., 2005*; *Yuan et al., 2015b*; *Gandasi et al., 2017*; *Ohara-Imaizumi et al., 2019a*), we observed a distinct distribution of secretion events to certain preferred areas of a cell (*Figure 2B–D*, *Figure 2—videos 1–3*). Analyzing the nearest neighbor distances between secretion events within cells, we detected a bias toward smaller distances, with over 50%

of events in all conditions occurring within 1.5 µm of each other (*Figure 3A*). We then performed a density-based scanning algorithm to identify clustering, defined as secretion events occurring within 1.5 µm diameter areas (*Figure 3B–D*, *Figure 3—figure supplement 1B-D*). To determine the expected frequency of clustered events occurring by chance (i.e., multiple unrelated events occurring close to each other by coincidence), we computationally simulated random secretion events in in silico cells. Results (*Figure 3—figure supplement 1A*) show that in cells secreting within the observed levels, clusters of two events within 1.5 µm of each other would be relatively prevalent (one or more per cell). In contrast, the likelihood of observing a cluster of three or more events due to random chance is very low. Thus, only cluster sizes of three and more events were included in the data presented below.

Our analysis (*Figure 3E and F*) revealed that the number of cells with secretion clusters was small under conditions with low secretion, including all low-glucose treatments and in high glucose after taxol pre-incubation. In contrast, the number of cells with clusters increased from on average 2% to 10% of the cell population by high glucose stimulation in control. Interestingly, nocodazole pretreatment increased the percentage of cells with clusters in high glucose 2.4-fold (*Figure 3E*). The nocodazole-induced increase in the number of cells secreting in a non-clustered manner was more modest (1.3-fold). This indicates that in the absence of MTs, a higher proportion of active cells had clustered secretion. Indeed, when only secreting cells were considered, the fraction of cells with clusters increased from on average 18% in control to 33% in nocodazole (*Figure 3F*). These data suggest that MT depolymerization specifically initiates secretion hot spots to activate dormant β-cells.

Interestingly, our correlative microscopy analyses of Glu-tubulin content in β-cells after secretion (as in *Figure 2F and G*) demonstrated that islet β-cells with high Glu-tubulin content (above the islet average) almost never had secretion clusters (*Figure 3G–J*), indicating that stable MTs in cells suppress hot spot-associated secretion.

## MTs regulate individual cell secretion via both clustered and non-clustered secretion

While our data indicated that the MT-dependent increase in secretion strongly relies on the activation of additional β-cells, we also detected that secretion activity per cell depended on MT presence and stability (*Figure 2I and J*). Exploring the secretion patterns in individual cells, striking differences between high-glucose stimulated control (*Figure 4A*, *Figure 4—video 1*) and nocodazole-treated (*Figure 4B*, *Figure 4—video 2*) cells are observed. We have found that, in addition to the significant increase in cluster-containing cells, MT depolymerization caused an increase in the number of secretion clusters per cell (*Figure 4C and D*), which, according to the computational simulation, cannot be accounted for by the increase in secretion per cell (*Figure 3—figure supplement 1A*). Interestingly, in a vast majority of cells with clustered secretion in control, the number of clusters per cell was restricted to one, while in nocodazole, many cells had additional clusters activated (*Figure 4E*).

In addition, while our data show that a significant number of exocytic events were specifically concentrated in secretion hot spots, many events were still found randomly scattered across the cell membrane. We tested how the number of non-clustered versus clustered events were changed upon glucose stimulation under conditions of MT depolymerization or stabilization. The number of clustered secretion events per cell increased between low and high glucose, and the loss of MTs resulted in an even higher increase (*Figure 4F*), reflecting the activation of additional hot spots as indicated above (*Figure 4C–E*). Interestingly, the non-clustered event number per secreting cell was not significantly increased by high glucose in control (*Figure 4F'*), indicating that GSIS is predominantly driven by clustered secretion. At the same time, glucose induced a consistent and significant rise in non-clustered secretion in nocodazole, indicating that without MTs, glucose-stimulated secretion can occur at random, non-hot spot sites (*Figure 4F'*).

## Individual events within clusters are independent of each other

To better assess how MT stability influences secretion dynamics within hot spots, we analyzed clustered secretion dynamics in comparison to non-clustered secretion. Our analysis detected a slight, statistically insignificant difference in the distribution of cluster sizes (the number of secretion events within a cluster) between conditions (*Figure 5A and A'*), suggesting that the level of secretion activity within these hot spots is to a large extent MT-independent.

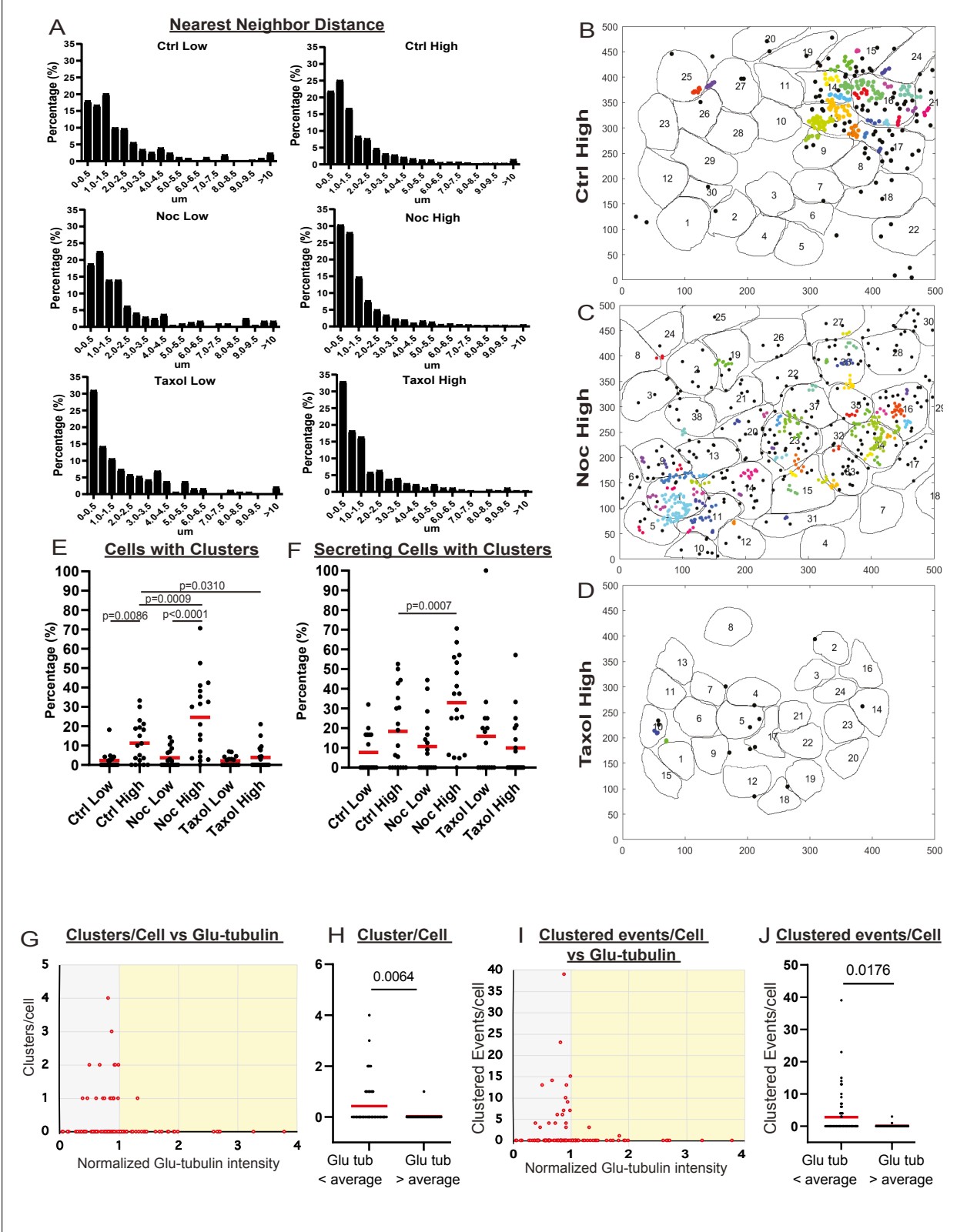

**Figure 3.** MT stability suppresses formation of insulin secretion hot spots. (**A**) Histogram of nearest neighbor distances obtained by measuring the distance between secretion events in cells with more than one secretion event during the movie. Graphed as the percentage of events within each bin per condition. Bin=0.5 μm. N=91–3255 distances from 16 to 19 islets. (**B–D**) Representative images of output from Matlab script (see Materials and methods) showing cell outlines (black lines) and secretion events (dots). Black dots are non-clustered secretion events, colored dots are clustered

*Figure 3 continued on next page*

*Figure 3 continued*

secretion events, each different color denotes a different cluster. Clusters were defined as 3+ secretion events occurring within nine pixels (1.44 µm) by density-based scanning. Islets were pre-treated with DMSO (control, **B**), nocodazole (**C**), or taxol (**D**) and were stimulated with 20 µm glucose. (**E**) Graph of the percentage of cells in each field of view with at least one cluster. Red bars, mean. One-way ANOVA and multiple comparison tests, p-value as indicated. N=16–19 islets. (**F**) Graph of the percentage of cells in each field of view with at least one cluster out of cells with at least one secretion event. Red bars, mean. One-way ANOVA and multiple comparison tests, p-value as indicated. N=16–19 islets. (**G**) Correlation of Glu-tubulin intensity (normalized to islet average) to the number of secretion clusters per cell in whole islets. Gray field, intensity below islet average (<1). Yellow field, intensity above islet average (>1). The same data set as in *Figure 2F–H*. (**H**) The number of clusters in cells with Glu-tubulin intensity below islet average and those above islet average is compared in the graph. Mann-Whitney nonparametric comparison test p-value is shown. N=98 cells from five islets. The same data set as in *Figure 2F–H*. (**I**) Correlation of Glu-tubulin intensity (normalized to islet average) to the number of clustered secretion events per cell in whole islets. Gray field, intensity below islet average (<1). Yellow field, intensity above islet average (>1). The same data set as in *Figure 2F–H*. (**J**) The number of clustered events in cells with Glu-tubulin intensity below islet average and those above islet average is compared in the graph. Mann-Whitney nonparametric comparison test p-value is shown. N=98 cells from five islets. The same data set as in *Figure 2F–H*. MT, microtubule.

The online version of this article includes the following source data and figure supplement(s) for figure 3:

**Source data 1.** Data for graphs depicted in *Figure 3A,E,F,G,H,J,I*.

**Source data 2.** Matlab script to identify secretion events and clusters.

**Figure supplement 1.** 3+ event clusters are not spurious and a very rare in basal glucose conditions.

We next assessed whether clustered and non-clustered secretion events are independent of each other. That is, does one secretion event influence the timing of the next (dependent) or not (independent). To test this, we analyzed the time between events, which if events are independent, should follow an exponential distribution. Non-clustered events are well fit by an exponential distribution (*Figure 5B*), suggesting that non-clustered events do not affect each other, as would be expected. A nonparametric Kolmogorov–Smirnov (KS)-test indicates that, in control and nocodazole treated islets stimulated with high glucose, the distribution does statistically deviate from exponential. However, this is likely due to the clear time dependence of secretion after glucose stimulation in these conditions (*Figure 6A*, further discussion below).

By comparison, the time-between-events distribution for clustered events clearly deviates from exponential (*Figure 5C*). This is particularly obvious under conditions of high glucose stimulation in control and nocodazole, where a high number of clustered events allows for constructing a smooth distribution. Specifically, there is an enrichment of short wait times between events that cannot be captured by the exponential. However, this analysis considers time-between-events within all clusters combined into a single distribution. In this case, the observed deviation may be caused by the presence of clusters with different secretion dynamics within analyzed cluster populations.

To test whether individual clusters exhibit distinct secretion dynamics, we next analyzed event dynamics separately in different cluster sizes. In control and nocodazole treated islets in high glucose (*Figure 5D*), the average time between events was lower for larger clusters. Individual analyses of the waiting time distributions for clusters of each size in each of the conditions (*Figure 5E*) indicate that those distributions are well fit by exponential distributions. Thus, for each condition, we constructed a generalized linear model (GLiM) with an exponential linking function (since the data are exponentially distributed) where the rate of secretion depends on cluster size according to (rate=$\alpha$+$\beta$*Size). This model was fit to the data for each of the six conditions separately using a Bayesian parameter estimation approach.

Our goal is to determine (1) whether ($\beta$>0) since this would indicate that there is a size dependence of cluster secretion rates and (2) whether the model parameters differ across conditions.

Results of these model fits are presented using 95% Bayesian credible intervals (*Kruschke and Liddell, 2018*; *Figure 5F*). In brief, a 95% credible interval depicts the range that the parameter values will fall into with 95% certainty (note this is technically different than confidence intervals that are often reported in frequentist statistics). Results show that the credible interval for $\beta$ is strictly greater than 0 in control and nocodazole treated islets in high glucose. Thus we can say that larger clusters do indeed secrete insulin at higher rates with 95% certainty (this is the definition of a credible interval, see Materials and methods for further technical discussion). This is confirmed by model comparison where this size-dependent model is compared to a 'null' model where the secretion rate is fixed across all sizes (model comparison results are reported using the Watanabe–Akaike information criterion – WAIC, which is a Bayesian companion of the common frequentist AIC). Results are inconclusive in the

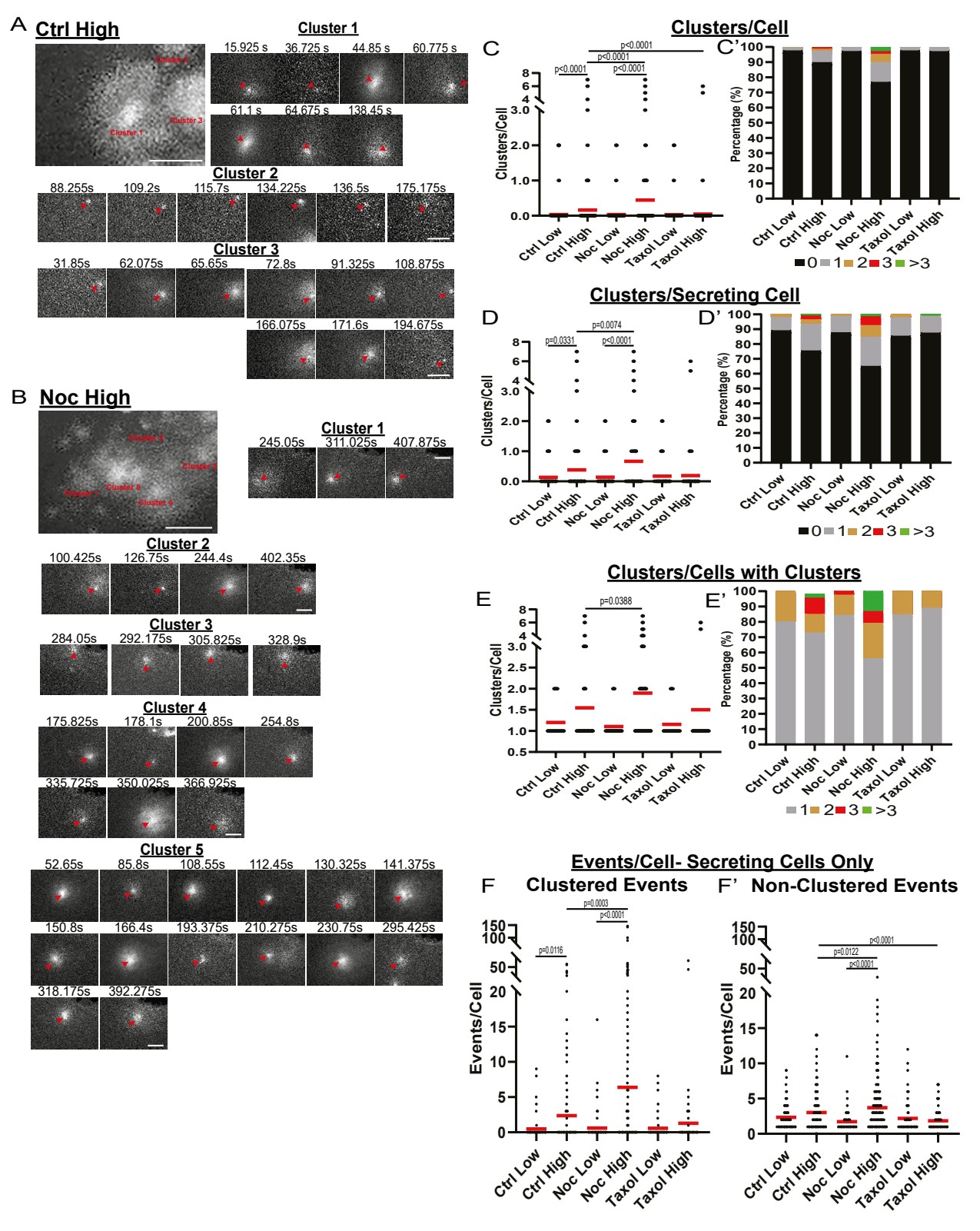

**Figure 4.** MT-disruption increases the number of hot spots per cell, increasing clustered secretion. (**A**) Representative images of clusters from one cell in a control islet stimulated with 20 mM glucose. Clusters were identified by Matlab script (see Materials and methods). First image is time projection through all clusters in the cell 15.925–194.675 s of the movie, clusters are identified by red text. Time in seconds of each event in the cluster above, red arrowheads identify the secretion event. Scale bars: 5 μm. (**B**) Representative images of clusters from one cell in a nocodazole pre-treated islet

*Figure 4 continued on next page*

*Figure 4 continued*

stimulated with 20 mM glucose. Clusters were identified by Matlab script (see Materials and methods). First image is time projection through all clusters in the cell 54.65–407.875 s of the movie, clusters are identified by red text. Time in seconds of each event in each cluster above, red arrowheads identify the secretion event. Scale bars: 5 µm. (**C**) Graph of clusters per cell in the field of view, with all cells whether activated during the movie or not. Red bar, mean. Kruskal-Wallis test nonparametric and multiple comparison tests, p-value as indicated, N=495–637 cells from 16 to 19 islets. (**C'**) Cells from panel (**C**), graphed as a stacked histogram of the percentage of total cells per condition that had each number of clusters. (**D**) Graph of clusters per cell only including cells with at least one secretion event during the duration of the movie. Red bar, mean. Kruskal-Wallis test nonparametric and multiple comparison tests, p-value as indicated, N=88–407 cells from 16 to 19 islets. (**D'**) Cells from panel (**D**), graphed as a stacked histogram of the percentage of cells with secretion events per condition that had each number of clusters. (**E**) Graph of clusters per cell, only cells with at least cluster during the duration of the movie were included. Red bar, mean. Kruskal-Wallis test nonparametric and multiple comparison tests, p-value as indicated, N=13–143 cells from 16 to 19 islets. (**E'**) Cells from panel (**E**), graphed as a stacked histogram of the percentage of cells with clusters per condition that had each number of clusters. (**F**) Graph of events per cell with at least one secretion event during the movie that were in a cluster. Red bar, mean. Kruskal-Wallis test nonparametric and multiple comparison tests, N=88–407 cells from 16 to 19 islets. (**F'**) Graph of events per cell with at least one secretion event during the movie that were not in a cluster. Red bar, mean. Kruskal-Wallis test nonparametric and multiple comparison tests, N=88–407 cells from 16 to 19 islets. MT, microtubule.

The online version of this article includes the following video and figure supplement(s) for figure 4:

**Source data 1.** Data for graphs depicted in *Figure 4C, C', D, D', E, E', F and F'*.

**Figure 4—video 1.** Clusters in a single cell in control islet in high glucose.

https://elifesciences.org/articles/59912/figures#fig4video1

**Figure 4—video 2.** Clusters in a single cell in nocodazole treated islet in high glucose.

https://elifesciences.org/articles/59912/figures#fig4video2

---

other four conditions due to the relative scarcity of clustering data leading to more widely distributed credible intervals that overlap with β=0. Comparing model results across conditions shows the credible intervals for both parameters (α not shown) overlap across all six conditions. Thus, there is no discernable difference in the rate of within-cluster secretion, consistent with our results in *Figure 5A*.

In conclusion, both clustered and non-clustered events appear to occur independently of each other: one secretion event does not influence the timing of the next. Neither cluster size nor secretion dynamics within clusters differ significantly between any of the examined experimental conditions. However, larger clusters are secreting insulin at faster rates.

## MT stability affects timing of glucose-stimulated insulin secretion

As insulin secretion is known to be tightly regulated in time, we tested if MT stability had any effect on the timing of secretion events. Since the data outlined above show that MT destabilization leads to the initiation of extra secretion hot spots, we also analyzed the kinetics of secretion for clustered and non-clustered events separately.

Events in low glucose occurred at random through time with no distinguishable pattern, indicating that basal secretion was random (*Figure 6A*). Control high glucose data showed a more noticeable pattern with the first peak at 2–4 min and the second peak at 9–10 min after high glucose addition (*Figure 6A*). Such kinetics correspond well with the known timing of the two waves of the bi-phasic insulin secretion (*Curry et al., 1975*). Interestingly, we found that clustered secretion at hot spots had a stronger impact on the first phase of secretion, while non-clustered events contributed stronger to the second phase (*Figure 6B*, *Figure 6—figure supplement 1*). In nocodazole-treated islets, high glucose induced a much broader peak of secretion with the maximum at 4–5 min, which extended into the timing of the second phase without a defined minimum (*Figure 6A*, *Figure 6—figure supplement 1*). Interestingly, the distribution of non-clustered events maintained a slightly bi-phasic appearance, while clustered secretion appeared as a single wide peak in the time frame of the experiment (*Figure 6B*, *Figure 6—figure supplement 1*). This indicates that the timing of clustered secretion was dysregulated.

To test whether the noted distribution is a manifestation of a change in the timing of triggering and/or silencing of secretion hot spots, we analyzed the distribution of the first and last events within each cluster over time (*Figure 6C and D*). This analysis showed that in control, clusters were initiated mostly in the first 5 minutes after stimulation (first phase), while in the absence of MTs clusters were initiated throughout most of the imaging sequence, suggesting that MTs serve to restrict cluster initiation to the first phase of GSIS (*Figure 6C and D*).

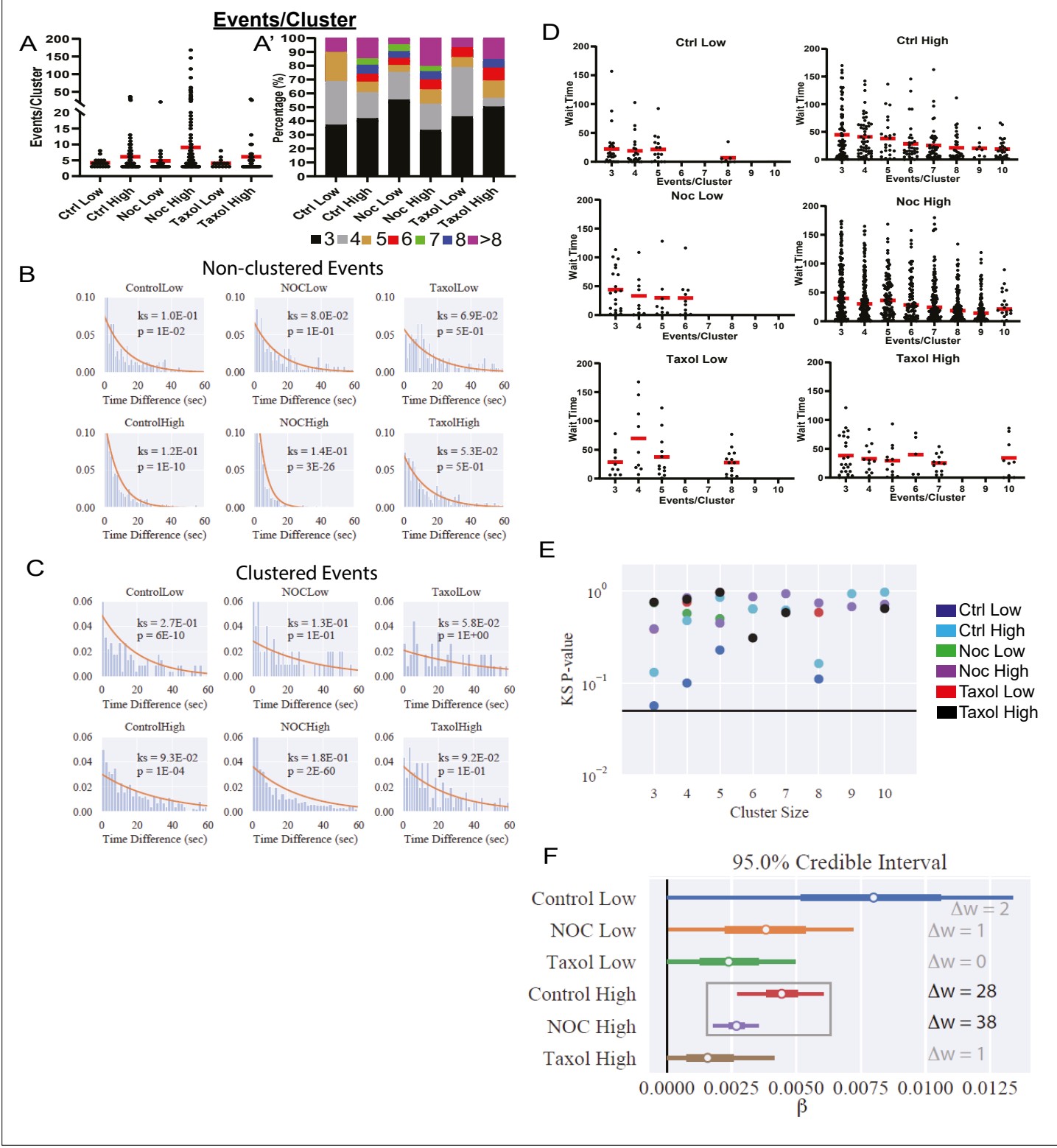

**Figure 5.** Increased secretion from clusters leads to faster secretion at that site. (**A**) Graph of events per cluster (cluster size), Red bar, mean. Kruskal-Wallis test nonparametric and multiple comparison tests found no statistical differences between conditions. N=14–290 clusters from 16 to 19 islets. (**A'**) Clusters from panel (**F**), graphed as a stacked histogram of the percentage of clusters with each number of events per condition. (**B**) Histogram of the time between successively (in time) occurring non-clustered events with the best fit exponential overlaid (KS-statistic is provided for quality of fit). (**C**) Histogram of the time between successively occurring clustered events with the best fit exponential overlaid (KS-statistic is provided for quality of

*Figure 5 continued on next page*

*Figure 5 continued*

fit). (**D**) Graph of time (seconds) between successive events (wait time). Distribution for clusters of different sizes (Red bar=mean). Some conditions lack clusters of particular size (e.g., no clusters with six secretion events in Ctrl low), as indicated by no data. (**E**) Each distribution in (**D**) is fit separately to an exponential distribution and the quality of fit is assessed with a KS-test (as in panels (**B, C**)). The resulting p-value for every test is plotted, with the black line indicating p=0.05. (**F**) Results of fitting a general linear model to the data from (**D**) (see Materials and methods for further details) with the assumption that 'secretion rate=α+β* Cluster Size.' Bayesian credible intervals for β are plotted for each condition. This model is also compared to a null model where 'secretion rate=α' (i.e., lacking size dependence), with model comparison results reported as the difference of WAIC scores (positive indicates the full model provides a better accounting of the data).

The online version of this article includes the following figure supplement(s) for figure 5:

**Source data 1.** Data for graph depicted in *Figure 5A*.

**Source data 2.** Python scripts that produce the statistical analysis and plots in *Figure 5B, C, D, E and F*.

Furthermore, in control glucose-stimulated islets, over 50% of clustered secretion ceased close to the end of the first phase, while the rest persisted through the second half of the movie. In the absence of MTs, in contrast, a noticeable proportion of clustered secretion lasted through to the end of the recorded sequence, and/or was still active during the last frame (*Figure 6C,D*). This indicates that while no difference in the cluster size between control and nocodazole (in any other conditions) was detected, timely cluster inactivation was disturbed in the absence of MTs, and secretion might continue at the same location for a longer period (not recorded due to the fluorescent dye background buildup).

These data are consistent with the model where MTs regulate the timely response of secretion hot spots to the stimulus, including both their initiation and silencing. Lack of such regulation led to 'smearing' of the bi-phasic GSIS response, so that the decrease in secretion after the first phase was not evident.

## Microtubules control insulin secretion in addition to, rather than via, Ca²⁺ signaling

Since MTs regulate secretion hot spots, which have been previously shown to be defined by $Ca^{2+}$ channel location at the plasma membrane (*Bokvist et al., 1995*; *Satin, 2000*), we have tested whether overall GSIS under conditions of MT disruption is still dependent on $Ca^{2+}$ influx. GSIS ELISA in intact islets has revealed that blocking $Ca^{2+}$ influx by diazoxide inhibits GSIS both in the presence and absence of nocodazole and eliminates secretion enhancement caused by MT disruption (*Figure 7A*). This indicates that $Ca^{2+}$-dependent pathway is essential for secretion stimulation regardless of MT presence/stability.

There is still a possibility that MT regulation may act upstream of $Ca^{2+}$ influx induction. To address this, we have tested whether pretreatment of islets with nocodazole influences glucose-dependent $Ca^{2+}$ influx in individual β-cells. We have utilized a fluorescent $Ca^{2+}$ reporter Calbryte 520 to follow single-cell response to glucose stimulation in attached islets in real time by confocal microscopy (*Figure 7B and C*, cyan, *Figure 7—videos 1; 2*). As above, β-cell identity was determined by the presence of nuclear mApple reporter (*Figure 7B and C*, red). No increase in $Ca^{2+}$ influx levels between control and nocodazole-treated cells was detected, arguing against the hypothesis of MT-dependent regulation of $Ca^{2+}$ influx (*Figure 7D and E*). While the response was expectedly variable between individual islets (*Figure 7—figure supplement 1*), single-cell $Ca^{2+}$ fluctuations in nocodazole treated cells (*Figure 7—figure supplement 1B*) show no detectable defect in timing and/or synchronization as compared to control, suggesting that both hub cell firing and cell connectivity was not affected by MT disrupture to a detectable extent. Interestingly, while no difference in summarized $Ca^{2+}$ influx over time has been detected (*Figure 7E*), our data indicate a slight decrease in the fluctuation amplitude (*Figure 7D*), suggesting that the absence of MTs might influence the functionality of $Ca^{2+}$ influx machinery in some minor way. Nevertheless, if such a defect exists, it must have a suppressing effect on insulin secretion, rather than underlie the observed GSIS enhancement.

Furthermore, we have explored whether single-cell secretion efficiency and heterogeneity is modulated by MTs under conditions of $Ca^{2+}$ influx enforced by 25 mM KCl (membrane depolarization). We have performed FluoZin-3 assays during KCl-induced insulin secretion. Because KCl causes immediate membrane depolarization, in these assays, we had to apply stimulation during TIRF-imaging recording in order to register early secretion events. In the time frame of the assay, disruption of

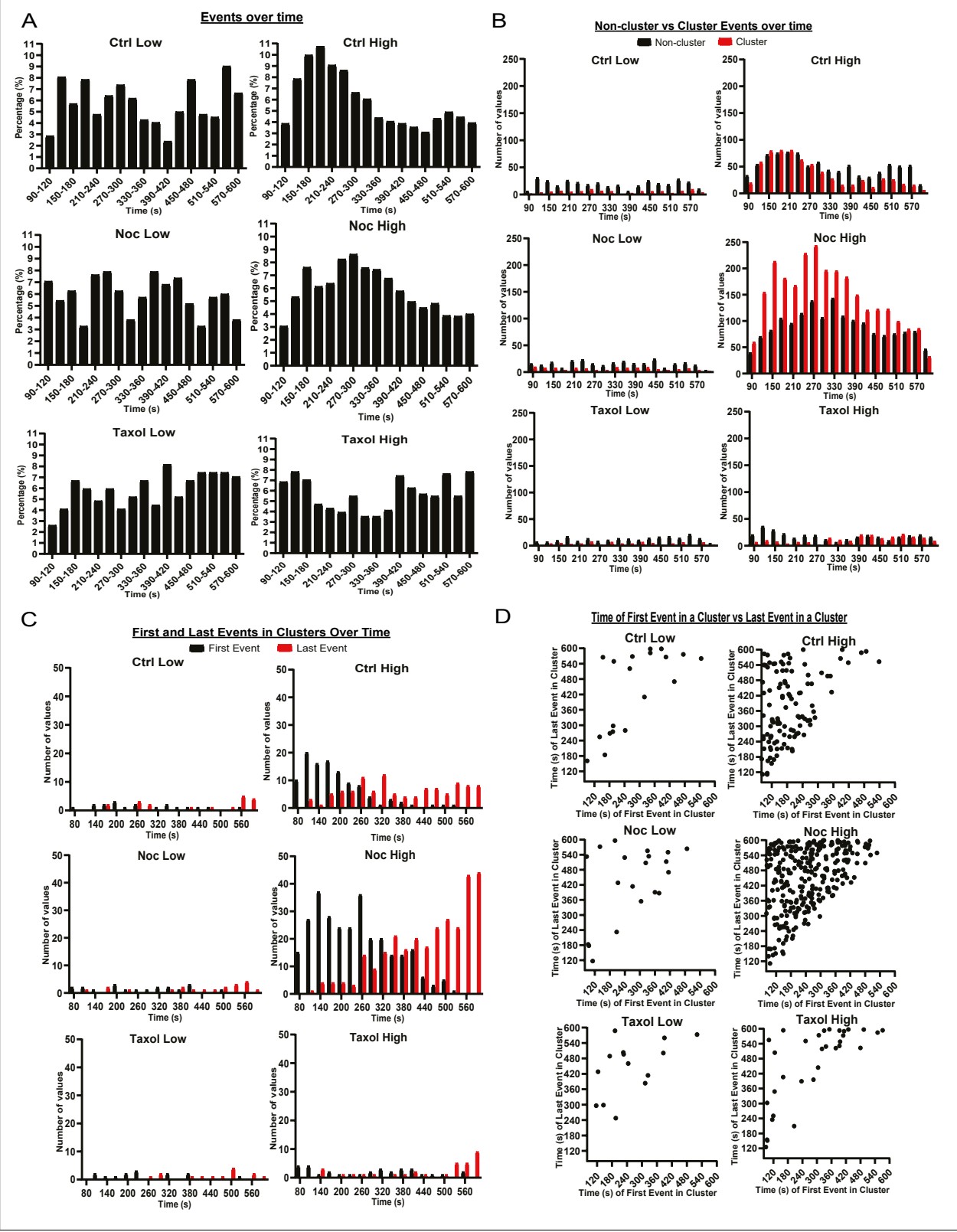

**Figure 6.** MTs restrict secretion from hot spots to the first phase of secretion, loss of MTs lengthen this phase. (**A**) Histogram of basal and glucose-stimulated secretion events over time. Graphed as the percentage of events within each bin per condition. Time (seconds) since dye and either high (20 mM) or low (2.8 mM) glucose addition. Bin=30 s. N=16–19 islets. (**B**) Histogram of secretion events over time separated into secretion events not in clusters (black) and in clusters (red). Absolute number of events is shown. Time (seconds) since dye and either high (20 mM) or low (2.8 mM) glucose

*Figure 6 continued on next page*

*Figure 6 continued*

addition. Bin=30 s. N=16–19 islets. (**C**) Histogram of the first event in a cluster (black) and last event in a cluster (red) over time. Absolute numer of events is shown. Time (seconds) since dye and either high (20 mM) or low (2.8 mM) glucose addition. Bin=30 s. N=16–19 islets. (**D**) Scatterplot of each cluster in each condition with the timing of the first event in a cluster on the x-axis and the timing of the last event in a cluster on the y-axis. Time (seconds) since dye and either high (20 mM) or low (2.8 mM) glucose addition. Bin=30 s. N=16–19 islets. MT, microtubule.

The online version of this article includes the following figure supplement(s) for figure 6:

**Source data 1.** Data for graphs depicted in *Figure 6A, B, C and D* and *Figure 1A*.

**Figure supplement 1.** Clustered secretion is mostly restricted to the first phase of secretion in control islets.

MTs by nocodazole pretreatment facilitated GSIS but did not affect secretion triggered by KCl alone (*Figure 8—figure supplement 1A*). However, synergistic secretion triggered by a combination of glucose and KCl was further enhanced under conditions of MT disruption. This effect was likely due to the increased number of secretion hot spots (clusters, *Figure 8—figure supplement 1B*), while the cluster size (number of events per cluster) remained constant throughout all conditions (*Figure 8—figure supplement 1C*).

At the same time, in the presence of KCl, secretion events within clusters occurred more rapidly (*Figure 8—figure supplement 1D*), and secretion concentrated within the first minutes after stimulation (*Figure 8—figure supplement 1E*), indicating that KCl-induced acute $Ca^{2+}$ influx acted predominantly in a short time frame. To analyze the role of MTs in secretion caused by KCl-dependent $Ca^{2+}$ influx, we have analyzed FluoZin-3 assay outcomes exclusively within the active period of 2.5 min (*Figure 8*). Strikingly, none of the components of secretion caused by KCl alone was facilitated by MT disruption (*Figure 8E–L*, *Figure 8—video 1*). However, secretion activity was significantly increased

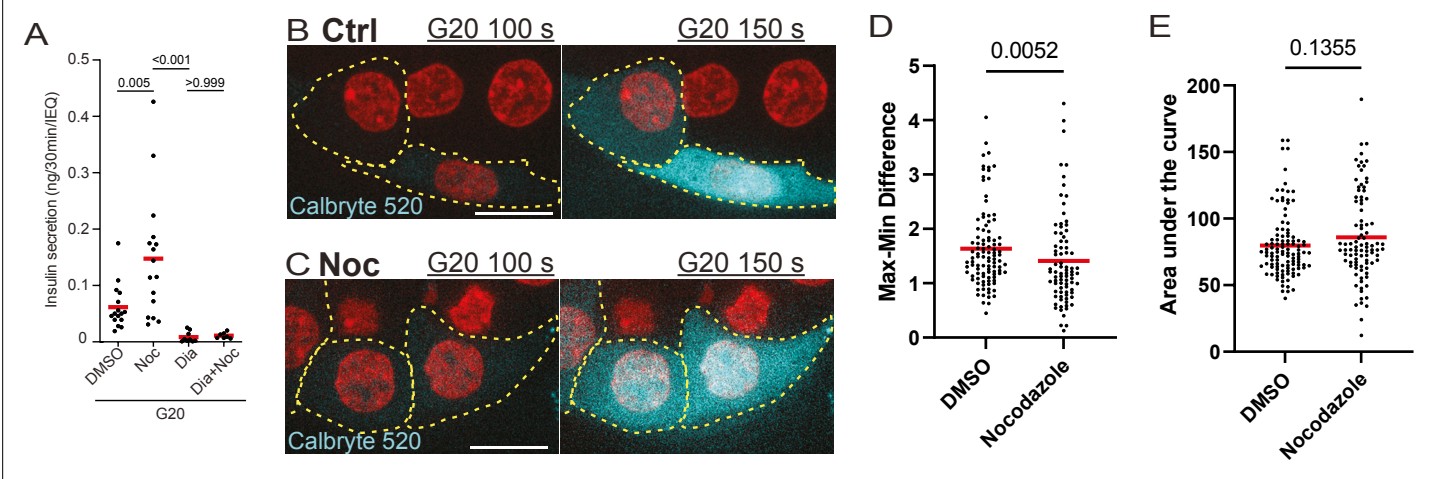

**Figure 7.** Calcium signaling is essential for secretion regardless of MT presence. (**A**) GSIS as detected by ELISA. Secretion over 30 min upon 20 mM glucose stimulation is shown for DMSO control and nocodazole pre-treated cells in the presence and absence of 5 μM Diazoxide. Tukey's multiple comparisons test p-values are shown. (**B, C**) Time frames from attached islets treated with DMSO (**B**) and nocodazole (**C**) and incubated with Calbryte 520 (cyan). Time after 20 MM glucose stimulation, seconds. Red, mApple (β-cell marker). Single plane spinning disk confocal microscopy images. Dotted line outlines indicate representative β-cells with detectable concentration of Calbryte 520 used for analyses. Scale bars: 10 μm. (**D**) Graph of highest amplitudes of Calbryte 520 intensity fluctuation per cell, measured in data as in *Figure 7—figure supplement 1*. Mann-Whitney nonparametric comparison test p-value is shown. N=78–102 cells from 8 to 12 islets. (**E**) Summarized increase of Calbryte 520 intensity over the first minute of glucose stimulation per cell, measured in data as in *Figure 7—figure supplement 1*. Mann-Whitney nonparametric comparison test p-value is shown. N=78–102 cells from 8 to 12 islets. GSIS, glucose-stimulated insulin secretion; MT, microtubule.

The online version of this article includes the following video and figure supplement(s) for figure 7:

**Source data 1.** Data for graphs depicted in *Figure 7A, D, E, Figure 7—figure supplement 1A, B*.

**Figure supplement 1.** Calcium influx over time in glucose-stimulated islets.

**Figure 7—video 1.** Glucose-dependent $Ca^{2+}$ influx in control detected by Calbryte 520.
https://elifesciences.org/articles/59912/figures#fig7video1

**Figure 7—video 2.** Glucose-dependent $Ca^{2+}$ influx in the absence of MTs detected by Calbryte 520.
https://elifesciences.org/articles/59912/figures#fig7video2

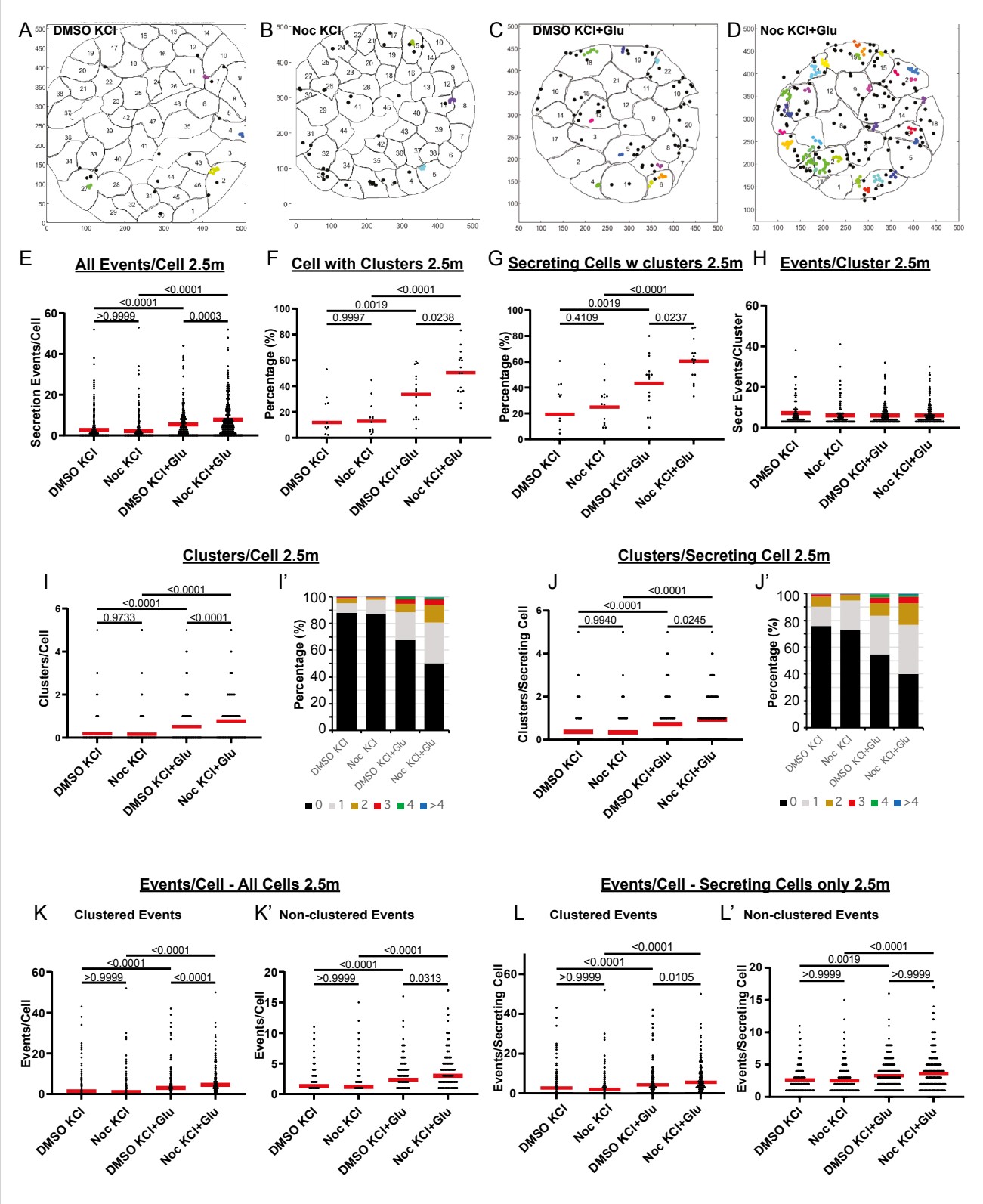

**Figure 8.** MTs regulate glucose-stimulated secretion in addition to calcium-dependent mechanisms. (**A–D**) KCl– (**A, B**) and KCl+ Glucose-stimulated (**C, D**) secretion as detected by FluoZin-3 assay in DMSO- (**A, C**) and nocodazole (**B, D**) pre-treated islets. Representative output images from Matlab script (see Materials and methods) shows cell outlines (black lines) and secretion events (dots) over 2.5 min after stimulation. Black dots are non-clustered secretion events, colored dots are clustered secretion events. (**E**) Graph of KCl-induced secretion events per cell over 2.5 min detected by FluoZin-3

*Figure 8 continued on next page*

*Figure 8 continued*

assay (data as in (**A–D**)). Red bar, mean. Kruskal-Wallis test nonparametric and multiple comparison tests, p-value as indicated, N=306–708 cells from 15 to 17 islets. (**F**) Graph of the percentage of cells in each field of view with at least one cluster in KCl-induced secretion over 2.5 min. Red bar, mean. One-way ANOVA and multiple comparison tests, p-value as indicated, N=15–17 islets. (**G**) Graph of the percentage of cells in each field of view with at least one cluster out of cells with at least one secretion event. KCl-induced secretion over 2.5 min. Red bar, mean. One-way ANOVA and multiple comparison tests, p-value as indicated, N=15–17 islets. (**H**) Number of secretion events per cluster (cluster size). KCl-induced secretion over 2.5 min. Red bar, mean. Kruskal-Wallis test nonparametric and multiple comparison tests, no significant difference, N=109–269 clusters from 306 to 708 cells/15–17 islets. (**I**) Graph of clusters per cell in each condition. KCl-induced secretion over 2.5 min. Red bar, mean. Kruskal-Wallis test nonparametric and multiple comparison tests, p-value as indicated, N=306–708 cells from 15 to 17 islets. (**I′**) Cells from panel (**I**), graphed as a stacked histogram of the percentage of total cells per condition that had each number of clusters. (**J**) Graph of clusters per cell only including cells with at least one secretion event. KCl-induced secretion over 2.5 min. Red bar, mean. Kruskal-Wallis test nonparametric and multiple comparison tests, p-values as indicated, N=218–334 cells from 15 to 17 islets. (**J′**) Cells from panel J, graphed as a stacked histogram of the percentage of cell with secretion events per condition that had each number of clusters. (**K**) Graph of clustered events per cell. KCl-induced secretion over 2.5 min. Red bar, mean. Kruskal-Wallis test non-parametric and multiple comparison tests, p values as indicated, N=306–708 cells from 15 to 17 islets. (**K′**) Graph of non-clustered events per cell. KCl-induced secretion over 2.5 min. Red bar, mean. Kruskal-Wallis test nonparametric and multiple comparison tests, p-values as indicated, N=306–708 cells from 15 to 17 islets. (**L**) Graph of clustered events per cell out of cells with at least one secretion event. Red bar, mean. Kruskal-Wallis test nonparametric and multiple comparison tests, p-values as indicated, N=218–334 cells from 15 to 17 islets. (**L′**) Graph of non-clustered events per cell out of cells with at least one secretion event. Red bar, mean. Kruskal-Wallis test nonparametric and multiple comparison tests, p-values as indicated, N=218–334 cells from 15 to 17 islets. MT, microtubule.

The online version of this article includes the following video, source data, and figure supplement(s) for figure 8:

**Source data 1.** Data for graphs depicted in *Figure 8E, F, G, H,I,I′,J,J′,K,K′,L,L′*.

**Figure supplement 1.** Rapid secretion induction by KCL as compared to glucose.

**Figure supplement 1—source data 1.** Data for graphs depicted in *Figure 8—figure supplement 1A, B, C, D, E*.

**Figure 8—video 1.** KCl-induced insulin secretion in DMSO and nocodazole.
https://elifesciences.org/articles/59912/figures#fig8video1

**Figure 8—video 2.** KCl- and glucose-induced insulin secretion in DMSO and nocodazole.
https://elifesciences.org/articles/59912/figures#fig8video2

**Figure 8—video 3.** KCl- and glucose-induced insulin secretion in DMSO and nocodazole.
https://elifesciences.org/articles/59912/figures#fig8video3

by nocodazole when glucose was used as a trigger in combination with KCl (*Figure 8E*, *Figure 8—videos 1–3*). The number of cells with clusters (*Figure 8F and G*) as well as the number of clusters per cell (*Figure 8I and J*) significantly increased upon nocodazole pretreatment, indicating an increased number of secretion hot spots. Accordingly, while the cluster size (number of events per cluster) remained constant (*Figure 8H*), the number of clustered events per cell was boosted by nocodazole (*Figure 8K and L*). The number of non-clustered events per cell was only slightly affected by nocodazole pretreatment (*Figure 8K′ and L′*), in agreement with our above conclusion that MTs predominantly regulate hop spot associated secretion.

Taken together, these data indicate that MT-dependent regulation of secretion hot spots acts in parallel to $Ca^{2+}$-dependent mechanisms and involves other pathways downstream of glucose metabolism.

## Discussion

It has been known for over six decades that islet β-cells, from both rodent models and human donar islets, can be divided into different subpopulations with different gene expression, morphology, and GSIS activities (*Miranda et al., 2021*). The functional β-cell heterogeneity, established via hemolytic plaque assays (*Salomon and Meda, 1986*; *Bosco et al., 1995*; *Katsuta et al., 2012*) or real-time secretion assays (*Li et al., 2011*; *Hoang Do and Thorn, 2015*) is particularly intriguing because it was proposed to be important for an adaptable, efficient response to various changes in physiological conditions and is one of the parameters that dramatically changes in diabetes (*Aguayo-Mazzucato et al., 2017*; *Gutierrez et al., 2017*). Mechanism-wise, it has been proposed that β-cell heterogeneity can result from differences in β-cell age, disease state, β-cell maturity, and location within the islet or association with other islet cell types (*Dean and Matthews, 1970*; *Efendić and Luft, 1975*; *Pipeleers et al., 1982*; *Stefan et al., 1987*; *Ballian and Brunicardi, 2007*;

*Wojtusciszyn et al., 2008*; *van der Meulen et al., 2015*; *Aguayo-Mazzucato et al., 2017*; *Gutierrez et al., 2017*; *Pipeleers et al., 2017*) and to vasculature (*Ballian and Brunicardi, 2007*; *Low et al., 2014*). Yet, these studies do not fully explain why similar levels of $Ca^{2+}$ influx cannot induce similar insulin secretion from different β-cells.

In this study, we have evaluated the role of the MT cytoskeleton in influencing the spatial distribution and heterogeneity of insulin secretion events both at the level of pancreatic β-cell populations and sub-plasma membrane regions. Our data indicate that MT stability in the β-cell population is heterogeneous and that this heterogeneity is a contributing factor to the previously established heterogeneity of insulin response. We also show that the loss of MTs causes initiation of additional insulin secretion: (1) activation of hot spots in a higher fraction of cells, (2) increase in the number of hot spots in active cells, and (3) broadens the timing of secretion from the hot spots in the first phase of GSIS. Stabilizing MTs prevents GSIS so that all parameters are indistinguishable from basal secretion levels. Importantly, MT-dependent regulation of hot spots acts in parallel to glucose-induced $Ca^{2+}$ influx, rather than in the same pathway.

There are a number of advantages to the approach utilized in this paper. It is important to measure the spatiotemporal distribution of secretion from individual β-cells in its natural environment in the whole islet with intact cell-to-cell contacts and a proper basement membrane. High-resolution detection of secretion event distribution over the surface of the plasma membrane is a challenging task. While secretion detection in whole islets has been successfully achieved by two-photon microscopy (*Takahashi et al., 2002*; *Low et al., 2014*), this method does not allow for high time resolution and for registration of secretion hot spots over the plasma membrane due to its complex topography. Furthermore, it is very labor-intensive and the number of examined samples is very limited. In contrast, TIRF microscopy with β-cells secreting toward the glass coverslip (*Nagamatsu and Ohara-Imaizumi, 2008*; *Loder et al., 2013*) allows for rapid imaging in the plasma membrane focal plane. The major challenge is the rapidly compromised responsiveness to glucose characteristic for whole-mount islets plated on glass. To our advantage, it has become clear that ECM signaling through integrin-dependent $Ca^{2+}$ channel activity is critical for the preservation of correctly patterned secretion (*Gan et al., 2018*; *Ohara-Imaizumi et al., 2019b*). Our approach of culturing whole-mount islets on vascular ECM preserves the functionality of β-cells (*Patterson et al., 2000*; *Zhu et al., 2015*), supporting the idea that the signals downstream of integrins are vital to preserving β-cell identity (*Gan et al., 2018*). TIRF microscopy in this system allows us to register single secretion events associated with their physiological organizer—the vascular ECM. Thus, this experimental model allows for the evaluation of the secretion patterning arranged by vascular cues, and at the same time, analysis of individual β-cell responses as they maintain their connections with each other and other islet cell types. An additional advantage of our approach is that utilization of FluoZin-3 dye provides direct information of precise insulin secretion time and location without the need for genetically encoded markers of insulin.

One important conclusion from our study is that MT stability varies in the β-cell population. The mechanistic basis of the differences in MT stability between β-cells is yet unclear. For example, since MTs are sensitive to $Ca^{2+}$ (*Hepler, 2016*), MT stability might be modified by the $Ca^{2+}$ influx wave, which is thought to contribute to spatial and temporal differences in β-cell response to stimulation (*Benninger et al., 2014*). However, we have recently shown that blocking $Ca^{2+}$ influx by verapamil does not prevent glucose-induced changes in MT dynamics (*Ho et al., 2020*). Keeping also in mind the evidence that $Ca^{2+}$ influx is insufficient to provide for secretion heterogeneity (*Li et al., 2011*), we suggest that MT differences are likely triggered by another mechanism. MT heterogeneity might be promoted by such factors involved in β-cell variability as cell maturity (*Aguayo-Mazzucato et al., 2017*; *Gutierrez et al., 2017*; *Pipeleers et al., 2017*), islet microenvironment (*Trimble and Renold, 1981*; *Trimble et al., 1982*; *Brereton et al., 2015*), and/or paracrine signaling from other islet cell types (*Efendić and Luft, 1975*; *Pipeleers et al., 1982*; *Wojtusciszyn et al., 2008*; *van der Meulen et al., 2015*). Interestingly, we have shown that glucokinase, an enzyme, which expression/activity is involved in promoting β-cell variability (*Jetton and Magnuson, 1992*, *Heimberg et al., 1993*), is important for MT network remodeling downstream of glucose (*Ho et al., 2020*). Thus, glucokinase variability could lead to heterogeneous MT stability in β-cells.

One possible important molecular player in this pathway could be the MT stabilizer tau, which we have recently found to be a critical component of glucose- and glucokinase-dependent MT

remodeling, leading to efficient insulin release from β-cells (*Ho et al., 2020*). Other MT-stabilizing proteins found in β-cells, such as MAP2 and doublecortin, may also be involved (*Krueger et al., 1997*; *Jiang et al., 2013*). It is clear from our data that regardless of the source of MT heterogeneity, it is functioning to enhance the variability of β-cell secretion activity: pushing MTs toward uniform stability or uniform depolymerization makes β-cell secretion response more uniform in opposite directions. This role of MT heterogeneity in the regulation of variable β-cell response to stimulation is not entirely unexpected. The effect of heterogeneity of MT stability in cell populations has been described in other cell types and is implicated in the variability of cell function. For example, motile cells that have directionally stable MT arrays have an increased ability for migration in wound healing assays (*Sugioka and Sawa, 2012*). Heterogeneity of MT dynamics in neurons has been found to underlie the distinction between the axon and dendrites (*Conde and Cáceres, 2009*), neurite capacity for cargo movement (*Franker and Hoogenraad, 2013*), and modulation of local signaling and rearrangements in neuronal connectivity (*Hoogenraad and Bradke, 2009*). Now, our data add β-cells to the list of cell types where the differential MT stability plays an important role in cell physiology.

How do differential MT dynamics promote differential secretion? The first possibility is the direct MT regulation of insulin secretion, when differently organized MT networks differentially transport insulin granules within individual cells. This aligns well with the model supported by our previous findings (*Zhu et al., 2015*; *Bracey et al., 2020*), where stable MTs act as tracks for the withdrawal of insulin granules from the plasma membrane, restricting secretion. Lack of this regulated withdrawal should allow for the increased secretion at random locations. Indeed, this is what we observed: the increase of glucose-stimulated non-clustered events when MTs are absent (nocodazole treatment). However, we found that MTs most prominently regulate heterogeneity of β-cells responsiveness to glucose specifically via activation of secretion clusters, or hot spots. Such specific restriction of clustered secretion would be achieved by MT-dependent withdrawal if secretion-competent granules are specifically accumulated near the hot spots prior to the secretion stimulus by an additional mechanism. Alternatively, specific MT withdrawal from the secretion hot spots could result from local molecular

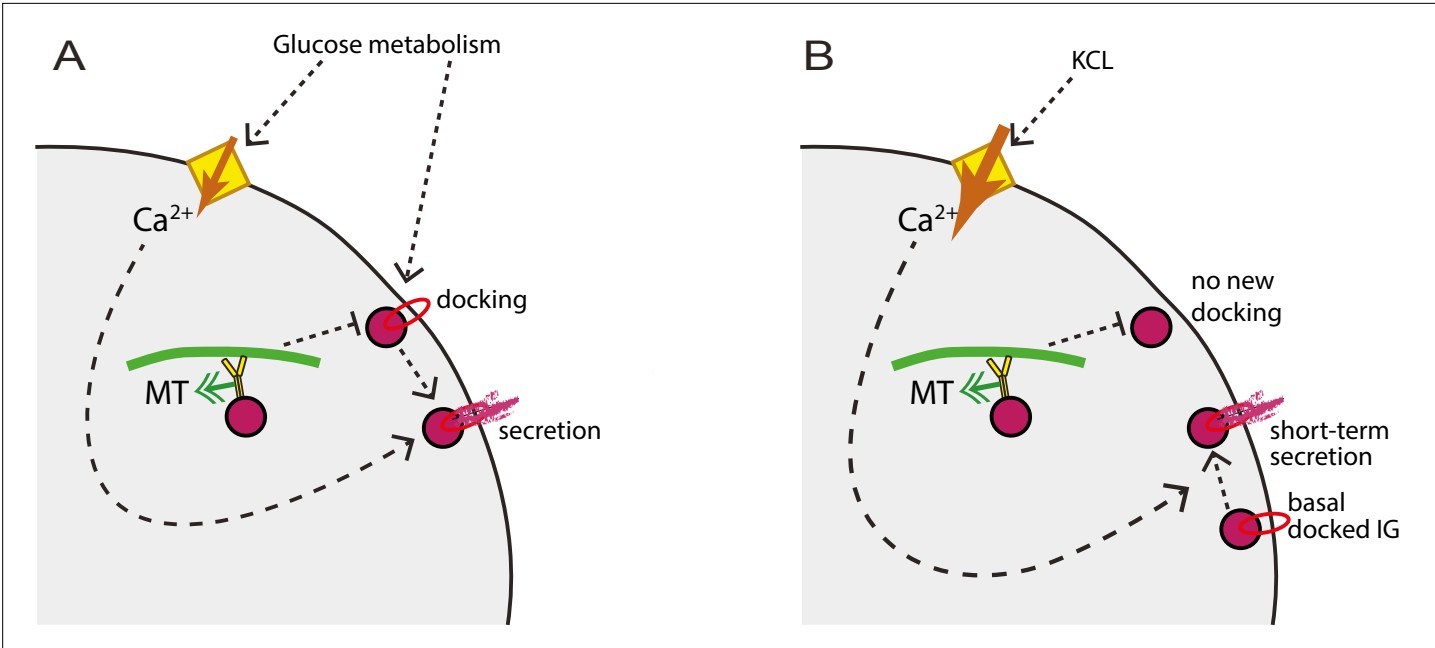

**Figure 9.** MT regulation of insulin secretion at a hot spot. (**A**) Glucose signaling stimulates both insulin granule docking and calcium influx; calcium influx, in turn, promotes secretion of docked graules. MT-dependent transport negatively regulates the process of docking, restricting the number of readily releasable granules and secretion outcome. (**B**) KCl treatment causes extreme acute calcium influx, which in turn facilitates secretion of pre-docked granules. No additional docking occurs in the absence of glucose stimulation, and MT regulation does not influence the number of pre-docked granules and secretion levels. MT, microtubule.

motor activation and/or local organization of MT tracks at these cellular locations. In this regard, it is interesting that ELKS, a tether molecule involved in MT anchoring at the cell cortex in some cell types (*Lansbergen et al., 2006*), is specifically concentrated at secretion hot spots. While ELKS is likely involved in the regulation of insulin exocytosis (*Ohara-Imaizumi et al., 2005*; *Ohara-Imaizumi et al., 2019b*), it might also be a regulator of local MT anchoring at those sites.

It is important to integrate our understanding of MT function with prior accumulated knowledge on secretion hot spots regulation. Existing evidence indicates that loci of repeated clustered secretion involve membrane accumulation of K$^+$ channels (*Fu et al., 2019*), VDCCs (*Bokvist et al., 1995*), as well as tethering and membrane fusion machinery (*Ohara-Imaizumi et al., 2004*; *Ohara-Imaizumi et al., 2007*). Accordingly, it is thought that the functioning of a hot spot includes insulin granule docking in preparation for secretion, ion channels to promote Ca$^{2+}$ influx, and Ca$^{2+}$-dependent membrane fusion. Our data indicate that MT depolymerization does not affect Ca$^{2+}$ influx, and that acute Ca$^{2+}$ influx stimulates secretion of readily releasable granules at hot spots regardless of MT presence, consistent with previous data (*Mourad et al., 2011*). Thus, Ca$^{2+}$-dependent steps of secretion are not MT-dependent. In contrast, our previous findings show that insulin granule docking at the plasma membrane is regulated by MTs, because MT-dependent transport withdraws granules from the docking sites (*Bracey et al., 2020*). We propose a model (*Figure 9A*) where glucose signaling stimulates both insulin granule docking and Ca$^{2+}$ influx, which are thought to be triggered via independent pathways (*Gaisano, 2017*). MT-dependent transport negatively regulates the process of docking, restricting the number of readily-releasable granules. Ca$^{2+}$ influx, in turn, promotes secretion of docked granules, in an MT-independent manner. In the experimental setup, when secretion is stimulated by KCl, only pre-docked granules are released, and MT disruption does not have a significant effect (*Figure 9B*).

An additional potential scenario is that dynamic MTs facilitate secretion at hot spots by coupling insulin exocytosis with compensatory endocytic events, as was shown for ELKS/LL5β patches in cultured Ins-1 cells (*Yuan et al., 2015a*). Another possibility is that MTs act indirectly by modulating one of the prominent mechanisms involved in β-cell heterogeneity regulation. Interestingly, such master regulators of heterogeneous β-cell response as gap junctions and ion channels have been shown to be modulated by MTs in other cell types (*Shaw et al., 2007*; *Joseph et al., 2014*).

Regardless of the mechanism, the important conclusion from our data is that it is the activation of hot spots in additional cells that makes the β-cell population response more uniform in the absence of MTs. Since Ca$^{2+}$ is essential but likely insufficient for β-cell heterogeneity (*Benninger and Hodson, 2018*), it is possible that suppression of hot spot activity by stable MTs provides a required layer of control restricting secretion in a subpopulation of cells in the presence of Ca$^{2+}$ influx. Furthermore, our finding that larger clusters secrete insulin at higher rates suggests a potential enhancement of secretion via a chemical (e.g., insulin signaling or ATP signals) or mechanical (e.g., in terms of membrane tension) feedback mechanism (*Rodríguez-Iturbe et al., 2002*; *Saltiel and Pessin, 2003*).

Another important take-home message from our analyses is that MTs regulate the timing of the first phase of GSIS via control of clustered secretion. We show that initiation of secretion hot spots that is normally rapidly triggered by the metabolic signal is dysregulated when MTs are lost, possibly due to the presence of excessive, unregulated readily-releasable granules. In the absence of MT regulation, new secretion sites continue to appear even at the time points when the first phase of secretion should normally be fading. This suggests a potential role for MTs in the decay of the first phase of GSIS, which could be a significant part of prevention of over secretion under physiological conditions. At the same time, MT-dependent regulation of random secretion applies to both stages of the bi-phasic GSIS. An extended first phase of secretion and enhanced secretion levels in both phases are consistent with our previous findings (*Zhu et al., 2015*).

At the same time, we do not find a significant role of MTs in the secretion dynamics within individual clusters. Our data clearly indicate that MTs do not affect the frequency of secretion events within clusters, which is at odds with a previous finding in KCl-stimulated Ins-1 cells plated on glass, where a decreased frequency of events within a cluster was observed in the absence of MTs (*Yuan et al., 2015b*). It is unclear at this point whether this discrepancy is due to the dramatic differences in the experimental models used in the two studies, but it is not surprising that the fine regulation detected here was not observed in the absence of islet cell interactions and vascular ECM signals.

To conclude, this study highlights a novel role of MT network in promoting β-cell heterogeneity in islets by restricting secretion hot spot activity.

# Materials and methods

## Key resources table

| Reagent type (species) or resource | Designation | Source or reference | Identifiers | Additional information |
|---|---|---|---|---|
| Genetic reagent (*Mus musculus*) | Ins-Apl mice | **Stancill et al., 2019** | | Histone 2B-mApple knocked into the *Ins2* locus |
| Cell line (*Homo sapiens*) | RPE1-hTert | ATCC, Cat# CRL-4000, | RRID:CVCL_4388 | |
| Antibody | Anti-$\alpha$-tubulin (DM1A clone) Mouse monoclonal | Sigma-Aldrich, Cat#: T9026 | | (1:500) dilution |
| Antibody | Anti-detyrosinated tubulin, rabbit polyclonal | Millipore, Cat#: AB3201 | | (1:500) dilution |
| Chemical compound, drug | FluoZin-3, Tetrapotassium Salt, cell impermeant | Thermo Fisher Scientific, Cat#: F24194 | | Final concentration (20 μM) |
| Peptide, recombinant protein | Human ECM | Corning, Cat#: 354237 | | Placenta-derived vascular ECM |

## Mouse utilization

Ins-Apl mice with Histone 2B-mApple knocked into the *Ins2* locus (**Stancill et al., 2019**) were used for all experiments. Males and females between 2 and 6 months were used. Data were separated by sex to determine statistical differences before being combined. Mice utilization was supervised by the Vanderbilt Institutional Animal Care and Use Committee (IACUC).

## Cell lines and maintenance

RPE1-hTert (ATCC) cells were maintained in DMEM/F12 with 10% fetal bovine serum (FBS) and antibiotic at 37°C with 5% CO2 and were periodically tested for mycoplasma.

## Islet picking, attachment, and dissemination

Mouse pancreatic islets were hand-picked following in situ collagenase perfusion and digestion. Islets were allowed to recover for at least 1 hr in RPMI 1640 Media (Life Technologies, Frederick, MD) supplemented with 10% FBS, 100 U/ml penicillin, and 0.1 cmg/ml streptomycin in 5% $CO_2$ at 37°C.

All coverslips and dishes were plasma cleaned and coated in placenta-derived human ECM (Corning, Cat#: 354237) which is comprised of laminin, collagen IV, and heparan sulfate proteoglycan, which serves as a reconstitution of vasculature ECM, for 30 min at 37°C.

For experiments utilizing attached intact islets, 5–8 islets per 10 mm glass bottomed dish (Mattek) were placed in the center of the glass in 100 μl RPMI 1640 and allowed to settle for 1.5–2 hr before being transferred to 5% $CO_2$ at 37°C. The following day, 900 μl of RPMI 1640 was added. Islet media were changed every 2–3 days for up to 9 days within which the experiment was performed. It has been previously shown that islets attached to vascular ECM preserve normal ability to secrete in response to glucose for up to 14 days (**Patterson et al., 2000**; **Zhu et al., 2015**). For experiments utilizing disseminated islets, 30–50 islets per coverslip or dish were used. Islets recovered for 4–24 hr following picking. To disseminate, all islets were collected into a 15-ml tube on ice and allowed to settle for 5 min. Most of the media were removed and islets were resuspended in 900 μl room temperature versene and mixed by pipetting up and down several times, and 100 μl warm 0.05% trypsin-EDTA was added. The mixture was pipetted up and down 25–30 times with a p1000 tip to allow for dissemination but keep clumps of islet cells present. Disseminated islets were then centrifuged at 300×*g* for 2 min at room temperature. The versene/trypsin solution was removed and cells were resuspended in 100 μl RPMI 1640 media per 30–50 islets. The cell mixture was added to coverslips or dishes.

For correlative microscopy after FluoZin-3 assays, dishes were scratched before plasma cleaning with a diamond pen in a pattern to aid in finding the same islets/cells after live cell imaging and immunostaining.

## FluoZin-3 Assay

FluoZin-3 assay was performed 4–9 days after picking to allow for robust attachement of the islets and the day following dissemination for disseminated islets. After 16–20 hr, RPE1-hTert cells expressing GFP were plated with islets at <5% confluency. The green signal in these cells is diffuse . These cells allow for more accurate determination of focus and TIRF angle for the assay as the green FluoZin-3 signal only appears once it is added to the dish.

On the day of the assay, islets were incubated at 37°C in low glucose (2.8 mM glucose) KRB (110 mM NaCl, 5 mM KCl, 1 mM MgSO4, 1 mM KH2PO4, 1 mM HEPES, 2.5 mM, CaCl2, and 1 mg/ml BSA) for 1.5–2 hr with a change of buffer after 1 hr. For nocodazole (Sigma-Aldrich, Cat#: M1404) treatment, stock solution was added to a final 5 µM concentration to treat islets for at least 4 hr before imaging. For taxol treatment (Sigma-Aldrich, Cat#: PHL89806), islets were treated for 2 hr before imaging with 5 µM taxol. Immediately before imaging, the buffer was replaced with 100 µl fresh buffer with the same treatments as before to reduce background.

Dishes were placed on the TIRF microscope and allowed to equilibrate for 10 min. An islet was identified by eye. A nearby RPE cell was used to focus the microscope to the bottom of the dish and set the TIRF angle of the green laser. A 10 µm stack of 0.2 µm slices was recorded of the islet before addition of the dye using both transmitted light and the 568 nm laser. Stacks were started below the islet to ensure the bottom of the cells were imaged. 50 µl of KRB buffer with the cell-impermeant FluoZin-3 dye (Thermo Fisher Scientific, Cat#: F24194) to final concentration of 20 µM was added. For high glucose treatment, glucose to a final concentration of 20 µM was added together with the dye. For KCl treatment, KCl to a final concentration of 25 mM was added together with the dye.

For 10 min assays, focus and TIRF angle were refined after dye addition and the recording (60 ms exposure, no delay) started within 2 min after glucose stimulation to register active GSIS. For the short-term assays (2.5 min), the stimulation and dye addition were performed during the recording (60 ms exposure, 100 ms/frame) to register rapid secretion in response to KCl. The maximal recording time was restricted to 8/10 min to avoid photo-toxicity.

## Processing of FluoZin-3 movies

Each image was subtracted from the previous image, briefly, the first and last frames from two different copies of the file were removed and subtracted using the Image Calculator tool in ImageJ with the 32-bit (float) result box checked. For 10 min of secretion analysis, every five images from this subtracted image were grouped as a max projection through time using the Grouped Z-project function in ImageJ and analyzed at 300 ms time resolution. For 2.5 image analyses, sequences were analyzed at 100 ms time resolution.

## Analysis of FluoZin-3 movies

### Identifying secretion events

Individual secretion events or 'flashes' were identified using ImageJ software (*Schindelin et al., 2012*). FluoZin-3 movies were blinded then analyzed manually. A macro was written that identifies the centroid of local maximal signals after the users use the point function identify a bright signal. Each event was identified as a flash of signal that was present in one frame only, with a noise level of 5000 and a search range of six pixels. To make sure that each single event was accounted for, an event was assigned to each evenly round spread of fluorescence. Rare non-round events were noted during 300 ms/frame analysis and re-analyzed in the 60 ms/frame sequence, where each secretion event is detected in more than one frame (*Figure 2A*). This analysis design supposedly resolved a vast majority of secretion events: it is highly improbable that subsequent events within a cluster are not resolved from each other, considering time resolution at 60 ms/frame or 100 ms/frame, XY spatial resolution at ~200 nm, and insulin granule size at ~300 nm in diameter. The outcome of analysis as the centroid coordinates were exported and used for analysis. On occasion, bright spots in the center of the event caused the formation of donuts preventing accurate identification of the centroid. These rare events were also noted during analysis and the centroid was identified manually and added to the data.

### Identifying cell borders

Transmitted light and 568 nm stacks were recorded before dye addition to identify individual β-cell borders. All cells with a red nucleus (Ins-Apl signal) were outlined by hand in ImageJ. If the nuclei

could not be seen (above the image stack range, signal diminished because of light dispersal or out of the frame) or was Ins-Apl negative, the cell was assumed to be a non-β-cell and discarded from analysis. Each Ins-Apl positive cell outline was saved as an individual ROI in ImageJ and coordinates were exported.

### Identifying secretion events/cell and clusters

A Matlab script (see supplemental annotated scripts) was used to compare the location of each secretion event (identified above) in relation to each cell (borders identified by ROI), outputting the number of secretion events in each cell. Some secretion events fell outside the boundaries of all β-cells identified and were discarded from analysis. Within the same annotated Matlab script clustering analysis was performed. Density-based scanning was used with a neighborhood search radius of 1.5 μm and a minimum number of neighbors of 3. The script outputs (1) the movie frame (time), centroid, cell, and cluster of each identified secretion event, (2) the cell number, secretion events, and clusters within each identified cell, and (3) the size of each cluster.

### GSIS assay

Isolated mouse islets were incubated in Krebs-Ringer bicarbonate buffer (KRB, 111 mM NaCl, 4.8 mM KCl, 25 mM NaHCO$_3$, 2.3 mM CaCl$_2$, 1.2 mM MgSO$_4$, 0.15 mM Na$_2$HPO$_4$, 1.2 mM KH$_2$PO$_4$, 10 mM HEPES, and 0.2% BSA) containing 2.8 mM glucose for 2 hr with 0.05% DMSO, 5 μg/ml Nocodazole, or 5 μM Diazoxide. Islets were then transferred to fresh KRB with the corresponding chemicals plus 2.8 mM or 20 mM glucose with or without 40 mM KCl and incubated for 30 min. The supernatant was collected and the insulin content was determined using the Mouse Ultrasensitive Insulin ELISA Kit (ALPCO, Salem, NH, Cat#: 80-INSMSU-E01). One islet equivalent (IEQ) is defined as a spherical islet with a diameter of 150 μm and is equal to 1,767,146 μm$^3$.

### Calbryte Ca$^{2+}$ influx assay and analysis

Mouse Islets (from Ins-Apl mice) were harvested and allowed to attach to human ECM (see above). On day 3 post-isolation (day of experiment), islets were preincubated for 2 hr in KRB with 2.8 mM glucose. 20 mM Calbryte 520 dye (AAT Bioquest, Cat#: 20650) was added 30 min prior to imaging to the final concentration 10 μM. For nocodazole treatment, the islets were preincubated with Nocodazole for 4 hr prior to imaging. The islets were imaged by spinning disk confocal microscopy at a single imaging plane at 50 ms exposure continuous imaging. Glucose was added to 20 μM final concentration as the onset of recording. For analysis, β-cells were identified by red nuclear marker. Calbryte intensity was measured within an ROI located inside individual cell borders excluding nucleus, and cells with average cytoplasmic intensity of at least 10 AU counts over background were analyzed. Because the individual cellular uptake of Calbryte dye varies, fold increase of intensity as compared to the time of stimulation was used for each cell. Images in *Figure 7* are single confocal slices.

### Immunofluorescence

#### Intact attached and disseminated islets following FluoZin-3 assay

Dishes were removed from the microscope stage and washed five times in PBS to remove the FluoZin-3 dye. Islets were fixed in 4% paraformaldehyde in PBS with 0.1% Triton-X 100% and 0.25% Glutaraldehyde for 1 hr(intact islets) or 10 min (disseminated islets) at room temperature. Following fixation, islets were washed five times in PBS with 0.1% Triton-X 100 at room temperature.

#### Disseminated islets for tubulin and Glu-tubulin intensity measurement

Dishes or coverslips were pre-treated with low glucose (2.8 mM) KRB at 37°C for 1.5–2 hr with a change in buffer after an hour. High glucose dishes or coverslips were then treated with high glucose (20 mM) KRB for 10 min. For ice treatment, the dishes or coverslips were placed on ice for 30 min before fixation. For cytosolic pre-extraction (dilution), cells were placed in 0.1% Triton in PEM buffer (0.1 M PIPES (pH 6.95), 2 mM EGTA, and 1 mM MgSO$_4$) for 1 min, then Triton was washed out, and islets were kept at 37°C in PEM buffer for 20 min before fixation. Cells were then fixed in 4% paraformaldehyde in PBS with 0.1% Triton-X 100% and 0.25% Glutaraldehyde. Following fixation, islets were washed five times in PBS with 0.1% Triton-X 100.

Primary and secondary antibodies were incubated for 24 hr (disseminated islets) or 48 hr (intact islets, antibody change after 24 hr). Samples were washed three times in PBS + 0.1% Triton-X 100 after primary and secondary antibody incubations. Following the final wash, post-fixing in 4% paraformaldehyde was performed (30 min for intact islets and 10 min for disseminated islets) and one more round of washing was performed before mounting coverslips.

Primary antibodies used are mouse anti-α-tubulin, DM1A clone (1:500, dilution, Sigma-Aldrich, Cat#: T9026), rabbit anti-detyrosinated tubulin (1:500, dilution, Millipore, Cat#: AB3201). Alexa488- and Alexa647-conjugated highly cross-absorbed secondary antibodies were obtained from Invitrogen. Coverslips were mounted in Vectashield Mounting Medium (Vector Labs, Cat#: H-1000).

## Microscopes

Fixed samples were imaged on a laser scanning confocal microscope Nikon A1r based on a TiE Motorized Inverted Microscope using a 100× lens, NA 1.49, run by NIS Elements C software. Cells were imaged in 2 μm slices through the whole cell for disseminated islets. Intact islets were imaged through 20 μm. Images in *Figure 1* are single slices from the bottom of the cells. Image in *Figure 1—figure supplement 1*, *Figure 2*, *Figure 2—figure supplement 1 Figure 2—figure supplement 2* are maximum intensity projections across three-ten slices from the bottom of the islet to better show the MT cytoskeleton.

FluoZin-3 assays for secretion analysis were imaged on a Nikon TE2000E microscope equipped with a Nikon TIRF2 System for TE2000 using a TIRF 100× 1.49 NA oil immersion lens and an Andor iXon EMCCD camera run by NIS Elements C software.

## Analysis of average intensity of tubulin and Glu-tubulin

β-cells were identified using red nuclei and outlined with each β-cell being assigned an ROI. The bottom of the cells were used for analysis as cell boaders were clearly visible and secretion in this location was measured in FluoZin-3 assays. Using ImageJ, the mean intensity of each cell in one slice at the bottom of the stack was measured. Both the Glu-tubulin and Tubulin channels were measured. Bright primary cilia signal was removed through thresholding of bright signals, assigning an ROI, and deleting the signal from both channels. A small box was drawn outside of the cell and measured in both channels for each image as the background. Background was then subtracted from the mean intensity within the cell.

## Statistical modeling methods

Given the distributional nature of this secretion event data, we use statistical modeling to both assess the properties of the process giving rise to the observed data and determine how those properties change under different conditions. A natural hypothesis to test is that secretion events occur independently of each other. If events are indeed independent, then the time between successive events within a cell or cluster should be exponentially distributed. To test this, we fit each data set using an exponential distribution using python's SciPy package. For non-cluster data, all events occurring within a cell were grouped in ascending order of occurrence time to produce a time-between-event distribution that was fit to an exponential distribution. For analysis of all cluster data within each condition (e.g., high glucose Taxol treated), events occurring within each cluster were grouped to compute the time between successive events within the same cluster. These wait times for all clusters were then grouped together to form a time-between-event distribution that was fit. This analysis however grouped clusters of different sizes together. We thus separately analyzed clusters of different sizes (measured as secretion events per cluster). For this, we collected the within cluster time between events for clusters of each size separately. We then grouped all computed waiting times for clusters of a given size (e.g., 4 secretion events) and fit the resulting distribution for each size separately to an exponential.

To assess whether clusters of different sizes secrete insulin at different rates, we constructed a GLiM where the secretion rate is linearly dependent on cluster size. Since the secretion time data for each cluster size is well approximated by an exponential distribution (as verified by the previously described analysis), we use an exponential linking function. Note that due to the non-normal nature of this data and the relatively small sample sizes, this GLiM approach is more appropriate than a more common ANOVA or similar approach. This GLiM was fit to the data for each condition separately

using Bayesian parameter estimation with the python PyMC package. Both the parameters ($\alpha$, $\beta$) were assigned half-normal priors with a standard deviation of 0.1, which are weakly informative priors. Four MCMC chains were used with 4000 samples each using the built in NUTS sampler.

We briefly discuss the interpretation of the Bayesian 'credible interval' approach we used for hypothesis testing on the parameter $\beta$. A typical frequentist approach to this would be to determine a 'point estimate' for the value of an unobserved parameter and construct a 'confidence interval.' Bayesian estimation instead produces a 'posterior density' (in contrast to a point estimate) describing the probability of every possible value of that unobserved parameter conditioned on the data. From this posterior, a 'credible interval' (Bayesian analog to a confidence interval) can be constructed. We specifically construct a 95% Highest Density Posterior Interval (HDPI), which is an interval within which 95% of the posterior density resides. This HDPI has a simple interpretation: given the observed data, there is a 95% probability that the parameter value falls within this interval. So if the 95% HDPI for $\beta$ does no overlap 0, we can be 95% certain that it is different from 0. Note that there are subtitle but important differences between credible intervals (Bayesian) and confidence intervals (frequentist) that are beyond the scope of this article. For further discussion, see *Kruschke and Liddell, 2018*.

## Experimental design

Glu-tubulin and tubulin intensity experiments were replicated at least three times. At least 20 images per experiment were obtained, random fields containing $\beta$-cells (identified by red nuclei) were imaged. All images where cell borders could be identified from the bottom of the stack where intensities were measured were analyzed.

Due to the large expected variation in insulin secretion from each $\beta$-cell (0 to >100 vesicles) reported by others, we aim to detect a 20% difference in means with a power of 0.95. This needs >327 samples (cells). FluoZin-3 assay experiments were repeated on at least 7 different days, using islets from independent isolations/mice. For all data sets, islets from the same animal/isolation were compared between two or more conditions. If at least one secretion event was identified, the movie was analyzed. Any movies without a secretion event were excluded as the technical difficulty of the assay caused an inability to identify a reason for lack of secretion. Analysis of secretion events was performed blind.

## Image processing

In order to make small structures visible, adjustments were made to brightness, contrast, and gamma settings of all fluorescent images presented here.

## Statistics

For data sets where the distribution was appropriate (normal distribution as determined by the D'Agostino & Pearson omnibus normality test), statistics were calculated by one-way ANOVA with Tukey's multiple comparisons test. When the data distribution was non-normal, a Kruskal-Wallis test nonparametric and multiple comparison was used (>2 data sets). For two data set comparisons, Mann-Whitney nonparametric comparison test was used. GraphPad Prism was used for statistical analyses and graphical representations. Significance was defined at p≤0.05.

## Acknowledgements

This work was supported by the National Institutes of Health (NIH) grants R01-DK106228 (to IK, GG, and WRH), R35-GM127098 (to IK), R01-DK125696 (to GG), DMS1562078 (to WRH), and R35-GM119552 (to MZ), T32 DK07061 and 1F32DK117529 (to KPT), and F31DK122650 (to KMB). MZ acknowledges support from the Searle Scholars Program. The authors thank Dr. David Jacobson for his advice on $Ca^{2+}$ imaging. The authors thank Hamida Ahmed, Dr. Alexey Khodjakov, and Dr. Anastasia Kaverina for technical help. The authors also thank Dr. Anneke Sanders for the critical reading of the manuscript.

## Additional information

### Funding

| Funder | Grant reference number | Author |
|---|---|---|
| National Institutes of Health | T32 DK07061 | Kathryn P Trogden |
| National Institutes of Health | 1F32DK117529 | Kathryn P Trogden |
| National Institutes of Health | R35-GM127098 | Irina Kaverina |
| National Institutes of Health | R01-DK65949 | Guoqiang Gu |
| National Institutes of Health | DMS1562078 | William R Holmes |
| National Institutes of Health | R01-DK106228 | Guoqiang Gu<br>William R Holmes<br>Irina Kaverina |
| National Institutes of Health | R35-GM119552 | Marija Zanic |
| National Institutes of Health | F31 DK122650 | Kai M Bracey |

The funders had no role in study design, data collection and interpretation, or the decision to submit the work for publication.

### Author contributions

Kathryn P Trogden, Conceptualization, Data curation, Formal analysis, Investigation, Methodology, Project administration, Resources, Validation, Visualization, Writing - original draft, Writing – review and editing; Justin Lee, Kai M Bracey, Kung-Hsien Ho, Formal analysis, Investigation; Hudson McKinney, Marija Zanic, Formal analysis, Software, Writing – review and editing; Xiaodong Zhu, Conceptualization, Methodology, Validation, Writing – review and editing; Goker Arpag, Thomas G Folland, Methodology, Software, Writing – review and editing; Anna B Osipovich, Resources; Mark A Magnuson, Resources, Writing – review and editing; Guoqiang Gu, Conceptualization, Funding acquisition, Methodology, Project administration, Supervision, Validation, Writing - original draft, Writing – review and editing; William R Holmes, Methodology, Software, Validation, Writing - original draft, Writing – review and editing; Irina Kaverina, Conceptualization, Formal analysis, Funding acquisition, Investigation, Methodology, Project administration, Software, Supervision, Validation, Writing - original draft, Writing – review and editing

### Author ORCIDs

Kathryn P Trogden (iD) http://orcid.org/0000-0003-3288-3859
Goker Arpag (iD) http://orcid.org/0000-0002-6893-2678
Marija Zanic (iD) http://orcid.org/0000-0002-5127-5819
Irina Kaverina (iD) http://orcid.org/0000-0002-4002-8599

### Ethics

This study was performed in strict accordance with the recommendations in the Guide for the Care and Use of Laboratory Animals of the National Institutes of Health. All of the animals were handled according to approved institutional animal care and use committee (IACUC) protocols (protocol M1500060-00) of Vanderbilt University.

### Decision letter and Author response

Decision letter https://doi.org/10.7554/eLife.59912.sa1
Author response https://doi.org/10.7554/eLife.59912.sa2

## Additional files

### Supplementary files
• Transparent reporting form

### Data availability
All numerical data generated during this study are included in the manuscript and supporting files. Source data files have been provided for all figures. Code is provided for computational data.

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
