## [Editor Report]

The study addresses the functional heterogeneity of glucose-stimulated insulin secretion by single pancreatic β cells. A focused analysis of spatiotemporal patterns of secretion in pancreatic islets led to the identification that cytoskeletal transport networks trigger secretion hot spots at the cell membrane and, in doing so, provide a layer of control over β cell heterogeneity. The study provides a yet uncharacterized dimension to the regulation of insulin secretion from the pancreas.

---

## [Decision Letter]

**Decision letter after peer review:**

Thank you for submitting your article "Microtubules regulate pancreatic β cell heterogeneity via spatiotemporal control of insulin secretion hot spots" for consideration by *eLife*. Your article has been reviewed by 3 peer reviewers, and the evaluation has been overseen by a Reviewing Editor and a Senior Editor. The following individuals involved in review of your submission have agreed to reveal their identity: David J Hodson (Reviewer #1); Jonathan S Bogan (Reviewer #2).

The reviewers have discussed the reviews with one another and the Reviewing Editor has drafted this decision to help you prepare a revised submission.

Summary:

The central finding is that microtubule stability is heterogenous in rodent β cells. Using Zn2+ reporter assays, the authors show that glu-tubulin/tubulin levels are associated with insulin secretion, and that modifying microtubule stability with nocodazole or taxol leads to changes in the proportion of active β cells, as well as insulin secretion dynamics at the single cell level. While microtubules are known to contribute to the regulation of insulin secretion, less is known about the role of microtubule stability in β cell heterogeneity. Thus, the authors' findings are highly novel. Furthermore, the analysis is well performed and the data will be of wide interest. Nocodazole and taxol are blunt tools, but the data are nonetheless informative and fit well with the authors' current work on effects of glucose-stimulated tau phosphorylation and destabilization of microtubules (Ho et al., Diabetes 2020). The Ho article slightly diminishes the novelty of this work, but a more important consideration is that it adds physiologic significance to this article. There are however a number of issues that need addressing, before an effect of microtubules on β cell heterogeneity can be firmly concluded. Specifically, the authors to give special attention to (a) studies on secretion in response to depolarization with KCl/diazoxide, (b) quantification of secretory granules and their dynamics, and (c) determining microtubule stability heterogeneity under a relevant condition mimicking T2D.

Essential revisions:

1. Experiments in models with perturbed islet dynamics would help to strengthen the main conclusion that differences in microtubule stability contribute to heterogeneity in insulin release. For example, the authors could compare distributions of glu-tubulin/tubulin/Fluozin3 in islets from HFD-fed mice.

2. The analysis of temporal and spatial aspects of exocytosis is nicely done, but it would be helpful to include data for the magnitude of each exocytic event in the analysis. Can the authors integrate the Fluozin-3 signal over time and space to estimate the relative size (or amount of Zn2+ that is secreted) in each event? This may be done using existing datasets. Is the size of a secretion event related to the likelihood that it is part of a cluster of exocytic events? One can imagine that compound exocytosis (which may increase the secretion at a single event) may be mechanistically similar to clustered exocytoses.

3. The data showing that nocodazole increased secretion only in the presence of high glucose, and not in low glucose, are interesting and imply that other factors are required for secretion. Possibly, ca^2+^ is one of these. Does KCl cause secretion in this setting, and does verapamil block secretion? Is glucose metabolism required? This is a critical experiment.

4. As acknowledged by the authors in the Discussion, changes to the microtubule network might conceivably affect glucose metabolism, calcium influx or mobilizsation and the study would be greatly strengthened by exploring at least one of these possibilities. To what extent, are calcium dynamics affected at the single cell level by nocodozole or taxol treatment in this preparation (experiments analogous to Figure 2)? Do those cells with greatest depolymerisation also experience the largest or most sustained calcium transients in response to glucose? On the same lines, it would be important to determine whether secretion (FluoZin3) responses to depolarization (e.g. KCl+diazoxide) were, like the response to high glucose, also heterogeneous, according to the extent of MT polymerisation? If there is also a correlation between the degree of depolymerization and KCl^-^stimulated secretion (at the single cell level) then this would provide good evidence that limitations in secretory granule dynamics (rather than in glucose metabolism or intracellular ca^2+^) directly influence exocytosis in a given cell and hence generate intercellular heterogeneity.

5. The use of permeablization to release free tubulin is the most direct measurement of microtubule stability, but this is shown only for high glucose treated cells. Was there a difference compared to low glucose? Can this be quantified?

6. The definition of insulin release clusters is not clear. At the resolutions used here (assuming ~100 nm in an ideal setup), it is unlikely that single release events can be reliably separated in a voxel < 300 nm. Thus, cluster events may in fact represent rapid sequential release of insulin granules in the same space. The authors should comment on this and what it means for a number of the measures presented herein.

7. How do insulin secretory hot spots correlate with glu-tubulin levels in the same cell? How can the increase in hot spots under conditions of microtubule re-arrangement (e.g. with nocodazole and taxol) be separated from changes in insulin tone and ergo paracrine feedback? A caveat might be needed here.

8. Data are obtained from multiple cells from multiple islets. However, are these islets from multiple islet isolations/animals? It is well known that individual islet isolation/animal is a bigger source of variability than individual islets.

9. Effects of nocodazole and taxol on glu-tubulin/tubulin fluorescence intensity distribution should be quantified (to show how reproducible these experiments are).

10. The flattened islet preparation is some way from an intact islet with native morphology, and further still from a fully vascularised and innervated islet in vivo. This limitation of the present study should be emphasised and the authors' claim that this preparation had to be used since TIRF microscopy is effectively the only way to measure secretion at the single cell level isn't convincing. Reports from Kasai (Takashi, N et al., Science, 2002) as well as that from Thorn (Low et al., Diabetologia, 2014) using alternative approaches (multiphoton microscopy of endocytosed probes such as sulphorhodamine) to measure exocytosis at the single cell level within the intact islet should be more clearly acknowledged. Rapid spinning disc confocal microscopy can also be used using recombinant targeted probes (Rutter et al., Biochem Soc Trans, 2006). These studies are not required, but the caveat needs discussion.

11. Figure 5. The complex data in this figure, obtained using Bayesian credible intervals, is difficult to understand, although the conclusion – that clusters do not influence each other – was clearly stated. This part of the manuscript needs clearer description and greater deconstruction of the mathematical underpinnings for the “typical” cell biologist.

---

## [Author Response]

Essential revisions:1. Experiments in models with perturbed islet dynamics would help to strengthen the main conclusion that differences in microtubule stability contribute to heterogeneity in insulin release. For example, the authors could compare distributions of glu-tubulin/tubulin/Fluozin3 in islets from HFD-fed mice.

We thank you reviewer for this valuable suggestion. However, such extension falls outside of the scope of this study. The goal of this paper is to establish the basic mechanisms as they exist in healthy β cells. We believe that our data justify the conclusion that differences in microtubule stability contribute to heterogeneity in insulin release in normal functioning islets. Thus, given the limited capacity of the lab during pandemics, and considering the significant expansion of the paper scope by other reviewers’ requests, we chose to leave this to our future studies.

2. The analysis of temporal and spatial aspects of exocytosis is nicely done, but it would be helpful to include data for the magnitude of each exocytic event in the analysis. Can the authors integrate the Fluozin-3 signal over time and space to estimate the relative size (or amount of Zn2+ that is secreted) in each event? This may be done using existing datasets. Is the size of a secretion event related to the likelihood that it is part of a cluster of exocytic events? One can imagine that compound exocytosis (which may increase the secretion at a single event) may be mechanistically similar to clustered xocytosis.

This definitely would be an interesting aspect to address. Unfortunately, integrated intensity of FluoZin signal quantified from those assays would not accurately represent the amount of secreted Zn2+ (which would correlate with insulin). In the TIRF assay, the brightness of the signal depends on how close it is to the interface between the glass coverslip and aqueous media (cells) within the TIRF illumination field (~200nm). Even a few nm difference in z-position of a secretion even would strongly influence the brightness of a flash. In an islet attached to a layered ECM, the distance between a β cell membrane and the coverslip is uneven, and it is impossible to interpret whether the flash intensity is due to the fluctuation of this distance or to the amount of secreted Zn2+. For this reason, we confidently interpret these data to determine only the XY positioning of the secretion events.

3. The data showing that nocodazole increased secretion only in the presence of high glucose, and not in low glucose, are interesting and imply that other factors are required for secretion. Possibly, ca^2+^ is one of these. Does KCl cause secretion in this setting, and does verapamil block secretion? Is glucose metabolism required? This is a critical experiment.

This is a very important point, and we thank the reviewer for bringing it up. We have previously shown that nocodazole does not cause secretion in the absence of glucose stimulus (Zhu et al., 2015), indicating that glucose metabolism is necessary for GSIS regardless of MT presence. Moreover, glucose-dependent MT remodeling triggered downstream of glucokinase (Ho et al., 2020). Now, we show that (1) Ca influx by KCl still induces secretion regardless of MT presence (new Figure 8); (2) blocking ca^2+^ influx by diazoxide inhibits GSIS both in the presence and absence of nocodazole (new Figure 7A); (3) glucose stimulation, in addition to calcium, is necessary for MT-dependent regulation of secretion (new Figure 8). This finding is in agreement with our hypothesis that MTs provide an additional layer of insulin secretion regulation under conditions of glucose stimulation. In other words, MTs are capable to suppress secretion at a subset of hotspots even when all Ca channels are open, while engagement of ca^2+^-dependent machinery is still a necessary requirement. We propose that glucose-activated MT transport removes insulin granules from some sites that otherwise – in MT absence – would actively dock and secrete upon stimulus (Figure 9).

4. As acknowledged by the authors in the Discussion, changes to the microtubule network might conceivably affect glucose metabolism, calcium influx or mobilizsation and the study would be greatly strengthened by exploring at least one of these possibilities. To what extent, are calcium dynamics affected at the single cell level by nocodozole or taxol treatment in this preparation (experiments analogous to Figure 2)?

To answer this comment, we have now detected Ca influx triggered by glucose using cell-permeable Ca sensor dye (Calbryte 520, new Figure 7B-D). No increase in Ca influx levels between control and nocodazole-treated cells was detected (new Figure 7C-D). Interestingly, Ca influx in both control and nocodazole-treated islet populations was similarly heterogeneous (new Figure 7—figure supplement 1). We conclude that MTs influence secretion and β cell heterogeneity independently of Ca, likely through a parallel mechanism(s). In addition, we observed a very slight decrease in Ca influx amplitude (new Figure 7C), suggesting that the absence of MTs might decrease functionality of Ca influx machinery in a minor way. Interpretation of this last observation is outside the scope of this study but we will follow it up in the future.

Do those cells with greatest depolymerisation also experience the largest or most sustained calcium transients in response to glucose?

We thank the reviewers for this refreshing question. We have not directly correlated the level of depolymerization to ca^2+^ activity at single β-cell levels. Such studies will be interesting but will be technically very challenging, because we will likely need to titrate the time course MT-destabilization and then record ca^2+^ dynamics simultaneously. But given that we detected no significant changes in glucose-induced ca^2+^ influx peak by complete MT depolymerization, this would be highly unlikely.

On the same lines, it would be important to determine whether secretion (FluoZin3) responses to depolarization (e.g. KCl+diazoxide) were, like the response to high glucose, also heterogeneous, according to the extent of MT polymerisation? If there is also a correlation between the degree of depolymerization and KCl^-^stimulated secretion (at the single cell level) then this would provide good evidence that limitations in secretory granule dynamics (rather than in glucose metabolism or intracellular ca^2+^) directly influence exocytosis in a given cell and hence generate intercellular heterogeneity.

Addressing this question, we have determined that secretion caused by KCl and KCl+glucose is highly heterogenous (Figure 8 and Figure 8 supplement). Destabilizing MT under the conditions of KCl only stimulation did not decrease heterogeneity, indicating that glucose-dependent but calcium independent processes are needed for MTs to be able to (de)activate hotspots. We can speculate that acute calcium influx induces secretion at already “assembled” hotspots, e.g. with a significant amount of already docked granules or “mature” pre-assembled docking machinery. When glucose signaling is also activated, hotspot “assembly” (docking, assembly of membrane molecular complexes) is facilitated in additional cells, decreasing heterogeneity, but is still partially suppressed by a MT-dependent mechanism (e.g. granule availability for docking). In this model, MT depolymerization would further decrease heterogeneity only under conditions of KCl+glucose stimulation, but not KCl by itself.

Interestingly, non-clustered secretion events were only marginally affected by nocodazole under the conditions of Ca influx+glucose, further emphasizing that, specifically, secretion at hotspots was a MT-regulated process.

5. The use of permeablization to release free tubulin is the most direct measurement of microtubule stability, but this is shown only for high glucose treated cells. Was there a difference compared to low glucose? Can this be quantified?

We agree that permeabilization to release free tubulin followed by MT staining/quantification is a good measure of microtubule dynamicity. Yet this is a damaging intervention and cannot exclude complications caused by the dilution of the cytoplasm. Thus, we include it only as an illustration. As an alternative, we have directly monitored MT depolymerization with live-time imaging of photolabeled preexisting MTs (published in Ho et al., 2020). We found that high glucose could substantially reduce MT stability in β cells of whole islets. Please refer to the original data (including detailed quantitative analysis) in this Ho et al., paper, which directly answer this question.

6. The definition of insulin release clusters is not clear. At the resolutions used here (assuming ~100 nm in an ideal setup), it is unlikely that single release events can be reliably separated in a voxel < 300 nm. Thus, cluster events may in fact represent rapid sequential release of insulin granules in the same space. The authors should comment on this and what it means for a number of the measures presented herein.

Please keep in mind that we detect clusters in 2D TIRF space. A single event is assigned to each resolved round spread of fluorescence. It is indeed possible that we do no resolve subsequent events that occur within 60 milliseconds (or 100 ms, Figures 7,8) and are separated by less than 200 nm. However, it is difficult to imagine that two 300nm granules move and perfectly align their centers in the same 200um area within 60 (100) msec. Moreover, our time resolution oftentimes allows us to follow the development of one event within two or three frames (see Figure 2A). Thus, while it is not impossible that we do not resolve some events, it deems highly improbable. For clarity, we now add a clarification of this point in the Methods section.

7. How do insulin secretory hot spots correlate with glu-tubulin levels in the same cell?

To address this question, we have now performed correlative microscopy experiments, when islets were fixed on the stage immediately after the FluoZin assay, stained for Glu-tubulin and re-imaged. We find a strong correlation between high Glu-tubulin content and inhibition of secretion (See new Figure 2 F-H, Figure 3G-J, and Figure 2—figure supplement 2). On the other hand, our data indicate that in cells with low levels of stable MTs, secretion heterogeneity is very high, suggesting a significant contribution of non-MT-dependent mechanisms. We would like to strike that we by no means intend to claim that cell heterogeneity is regulated by MTs only, and we are looking forward to future insights combining MT regulation with such things as paracrine signaling.

How can the increase in hot spots under conditions of microtubule re-arrangement (e.g. with nocodazole and taxol) be separated from changes in insulin tone and ergo paracrine feedback? A caveat might be needed here.

We thank the reviewer for this stimulating question. We agree that the secreted insulin and/or other co-secreted molecules or even the vesicle-plasma membrane fusion per se could provide chemical (e.g., insulin signaling or ATP signals) or mechanical (e.g., in terms of membrane tension) feedbacks, further modulating microtubule-dependent regulation of hot spots. It is consistent with our finding that “larger clusters do indeed secrete insulin at higher rates”. This perspective is now added in the discussion and will be extensively explored in the future studies.

8. Data are obtained from multiple cells from multiple islets. However, are these islets from multiple islet isolations/animals? It is well known that individual islet isolation/animal is a bigger source of variability than individual islets.

Yes, absolutely. Each experiment was repeated on several days using islets from separate isolations. Islets from the same animal/isolation were compared between two or more conditions. We now emphasize this information in the Methods section (“Experimental Design”).

9. Effects of nocodazole and taxol on glu-tubulin/tubulin fluorescence intensity distribution should be quantified (to show how reproducible these experiments are).

In nocodazole, there are no polymerized MTs, thus there would be no Glu-tubulin present to be quantified. This effect as well as effects of taxol causing increased Glu-tubulin levels have been are established for many decades, (Gundersen, Khawaja et al., 1987, Wehland and Weber 1987, Lin, Gundersen et al., 2002, Townley, Zheng et al., 2015, Baas, Rao et al., 2016, Gobrecht, Andreadaki et al., 2016) We have now included references to some of this literature to the manuscript.

10. The flattened islet preparation is some way from an intact islet with native morphology, and further still from a fully vascularised and innervated islet in vivo. This limitation of the present study should be emphasised

It was previously shown by David Piston’s lab and us that islets attached to placenta-derived ECM preserve their normal responsiveness to glucose for up to 14 days. The advantage of our method is that it is a model of secretion distribution within 2 dimensions of plasma membrane contacting vascular ECM (a model of cell area contacting capillaries). Thus, while these islets are ex-vivo, and our approach is more of a model, we don’t see it as a caveat, because it serves a different purpose of high time- and space-resolution detection of secretion loci on the cell surface toward the “modelled vasculature”.

It is also important to keep in mind that so-called “flattened” islets are not flat; rather, they are 3-dimentional intact islets, which have developed a flat area of vascular ECM attachment, the region that we are examining. We have now reworded the description of the method to make these point clear, and removed a misleading word “flattened”.

And the authors’ claim that this preparation had to be used since TIRF microscopy is effectively the only way to measure secretion at the single cell level isn’t convincing. Reports from Kasai (Takashi, N et al., Science, 2002) as well as that from Thorn (Low et al., Diabetologia, 2014) using alternative approaches (multiphoton microscopy of endocytosed probes such as sulphorhodamine) to measure exocytosis at the single cell level within the intact islet should be more clearly acknowledged. Rapid spinning disc confocal microscopy can also be used using recombinant targeted probes (Rutter et al., Biochem Soc Trans, 2006). These studies are not required, but the caveat needs discussion.

We thank the reviewers for discussing these variable techniques. Each of the studies referred to here are quite insightful, and the usage of techniques is suitable for questions asked in those studies. However, unlike TIRF microscopy, these approaches are not usable to address the questions that we are exploring in our manuscript.

Spinning disk confocal microscopy, which was used for follow NPY secretion from Min6 cells in (Rutter et al., 2006), is not suitable for the studies in 3D tissues or explants, such as islets, because confocality is lost due to out-of-focus light detected through extra pinholes. Furthermore, the usage of genetically encoded probes has its own set of pitfalls and is very challenging to prove that such markers highlight every insulin granule (in most cases they don’t).

By detecting non-specific dye penetration into exocytosed granules during their fusion with the membrane, Takahashi et al., and Low et al., were able to detect positions of single secretion events in isolated islets using two-photon microscopy. This approach is quite advanced and allows for examining secretion events inside the islet: the authors are able to compare the areas on capillary contact versus non-vessel-contacting cell surface. But while this technique has led to exciting findings, it is impossible to analyze secretion at high time resolution in three dimensions. The analysis in those seminal studies is done at single focal planes with secretion events distributed along the cell border in one optical section (in 1 dimension), or visualized in 3D at low time resolution. Thus, it does not allow for registration of secretion hot spots over the extended surface of a cell. Furthermore, it is very labor-intensive and the number of examined samples is very limited.

Thus, we believe that our chosen technique is particularly suited to our study, while those other techniques were advantageous is studies where they were used.

We have now added considerations of the advantages of our method to the manuscript.

11. Figure 5. The complex data in this figure, obtained using Bayesian credible intervals, is difficult to understand, although the conclusion – that clusters do not influence each other – was clearly stated. This part of the manuscript needs clearer description and greater deconstruction of the mathematical underpinnings for the "typical" cell biologist.

We have updated the relevant results text to help clarify the intent and conclusion in this analysis. The main idea is that a 95% credible interval is an interval in which you are 95% certain the relevant parameter value will fall. If the 95% credible interval for b is exclusively greater than 0, then we can be 95% certain that b > 0. To (slightly) more technically describe a Bayesian credible interval and its interpretation, we have added an additional paragraph to the modeling methods. Further discussion would require more in-depth discussion of Bayesian statistics, which is beyond the scope of these methods.